# *HIF-1α* is required for disturbed flow-induced metabolic reprogramming in human and porcine vascular endothelium

David Wu[1], Ru-Ting Huang[1], Robert B Hamanaka[1], Matt Krause[1], Myung-Jin Oh[1], Cheng-Hsiang Kuo[1], Recep Nigdelioglu[1], Angelo Y Meliton[1], Leah Witt[1], Guohao Dai[2], Mete Civelek[3], Nanduri R Prabhakar[4], Yun Fang[1*†], Gökhan M Mutlu[1*†]

[1]Department of Medicine, Section of Pulmonary and Critical Care Medicine, The University of Chicago, Chicago, United States; [2]Department of Bioengineering, Northeastern University, Boston, United States; [3]Department of Biomedical Engineering, University of Virginia, Charlottesville, United States; [4]Institute for Integrative Physiology, The University of Chicago, Chicago, United States

**Abstract** Hemodynamic forces regulate vascular functions. Disturbed flow (DF) occurs in arterial bifurcations and curvatures, activates endothelial cells (ECs), and results in vascular inflammation and ultimately atherosclerosis. However, how DF alters EC metabolism, and whether resulting metabolic changes induce EC activation, is unknown. Using transcriptomics and bioenergetic analysis, we discovered that DF induces glycolysis and reduces mitochondrial respiratory capacity in human aortic ECs. DF-induced metabolic reprogramming required hypoxia inducible factor-1α (*HIF-1α*), downstream of NAD(P)H oxidase-4 (*NOX4*)-derived reactive oxygen species (ROS). *HIF-1α* increased glycolytic enzymes and pyruvate dehydrogenase kinase-1 (*PDK-1*), which reduces mitochondrial respiratory capacity. Swine aortic arch endothelia exhibited elevated ROS, *NOX4*, *HIF-1α*, and glycolytic enzyme and *PDK1* expression, suggesting that DF leads to metabolic reprogramming in vivo. Inhibition of glycolysis reduced inflammation suggesting a causal relationship between flow-induced metabolic changes and EC activation. These findings highlight a previously uncharacterized role for flow-induced metabolic reprogramming and inflammation in ECs.

*For correspondence: yfang1@medicine.bsd.uchicago.edu (YF); gmutlu@medicine.bsd.uchicago.edu (GMM)

†These authors contributed equally to this work

Competing interests: The authors declare that no competing interests exist.

## Introduction

Atherosclerotic cardiovascular disease remains the leading cause of morbidity and mortality in the United States (*National Heart, Lung, and Blood Institute, 2013*). Altered blood flow characteristics (unidirectional vs. disturbed flow) and associated changes in flow-generated mechanical forces (hemodynamics) play a critical role in the focal nature of atherosclerosis (*Davies et al., 2013*; *Gimbrone and García-Cardeña, 2016*; *Zhou et al., 2014*). Unidirectional blood flow (UF) is associated with high time-averaged shear stress and is 'athero-protective' promoting a healthy endothelium, characterized by endothelial cell quiescence and maintenance of vascular barrier integrity. In contrast, disturbed flow (DF), which occurs in areas where atherosclerosis develops, is associated with low time-averaged shear stress and is 'athero-susceptible' promoting EC 'activation' characterized by inflammation and reduced vascular barrier integrity (*Davies et al., 2013*; *Zhou et al., 2014*; *Dai et al., 2004*; *Chiu and Chien, 2011*; *Liao, 2013*; *Abe and Berk, 2014*; *Xiao et al., 2013*). These areas where ECs are subjected to DF are the branched and curved parts of the arterial tree and underscores the focal, flow-dependent nature of endothelial 'activation'. These flow-dependent

**eLife digest** Atherosclerosis is the build-up of fatty material inside the blood vessels, and is one of the leading causes of heart disease and stroke. The blood vessels affected are typically inflamed for many years before the condition develops, and the condition often occurs at sites where blood vessels branch or turn.

The cells that line the inside of the blood vessels are known as endothelial cells. Flowing blood exerts a force upon the endothelial cells, named "shear force", which is similar to how wind bends plants. When the blood flows in one direction, the shear forces are high, the endothelial cells are tightly held together, and the vessels are less likely to become inflamed. However, the flow of blood is disturbed around turns or branch points. This means thatthe shear forces are lower and that the gaps between the endothelial cells are bigger. Low shear forces also mean that the endothelial cells release chemical signals that promote the inflammation and ultimately leads to atherosclerosis. Though low shear forces play an important role in "activating" endothelial cells to promote inflammation, it was not clear how this happens.

Wu et al. now show that when shear forces inside blood vessels are low, endothelial cells promote inflammation by modifying their own metabolism. The experiments involved applying either high or low shear forces to endothelial cells that had originally been collected from a major blood vessel of human donors, and then grown in the laboratory. Wu et al. then analyzed the gene activity of these endothelial cells and discovered that low shear forces activate a selected pool of genes. The activated genes are mainly responsible for two cellular processes: glycolysis and the response to hypoxia. Glycolysis is a process that releases energy by breaking down the sugar glucose, while hypoxia refers to the situation when cells do not receive enough oxygen. Further molecular analyses revealed that low shear forces stabilize a particular protein involved in the response to hypoxia, named HIF-1α, and that this protein is responsible for stimulating glycolysis. Finally, Wu et al. showed that increasing glycolysis in endothelial cells was enough to cause the blood vessels to become inflamed.

Going forward, a better understanding of how low shear forces modify the metabolism of endothelial cells in blood vessels and consequently promote inflammation will help scientists to tackle new questions about how atherosclerosis begins and develops. In the longer-term, these findings might also lead to the development of new treatments to atherosclerosis and similar diseases.

harbingers of atherosclerosis develop before any visible signs of disease (*Hajra et al., 2000*; *Won et al., 2007*).

Endothelial cells are plastic and their phenotypes are tightly regulated by hemodynamic changes. When cultured under static conditions, ECs exhibit increased expression of pro-inflammatory cyto-kines, reduced nitric oxide production (*Kizhakekuttu et al., 2012*; *Chen et al., 2010*; *Eelen et al., 2015*), and a shift towards glycolysis (*Eelen et al., 2015*; *De Bock et al., 2013a*). Compared to static conditions, UF reduces glycolysis in a Krüppel-like Factor 2 (*KLF2*)-dependent manner (*Dekker et al., 2002*; *Doddaballapur et al., 2015*) and increases mitochondrial biogenesis (*Chen et al., 2010*; *Kim et al., 2014*) but whether UF alters mitochondrial function is controversial (*Doddaballapur et al., 2015*). Furthermore, the effect of physiological DF, as opposed to no flow, on cellular metabolism is unclear and the mechanisms behind these DF-induced changes remain under-explored (*De Bock et al., 2013a*; *Doddaballapur et al., 2015*; *Cucullo et al., 2011*; *Wilhelm et al., 2016*).

Here, we employed RNA-seq, pathway analyses and complementary in vitro and in vivo systems to study the effects of DF and UF on cellular metabolism of human aortic endothelial cells (HAECs). We show that DF induces a metabolic phenotype of increased glycolysis and reduced oxidative phosphorylation, compared to UF. Our RNA-seq analyses predicted *HIF-1α* as a major transcriptional regulator controlling the DF-induced changes in endothelial phenotypes. Mechanistically, we demonstrate that DF activates hypoxia-inducible factor-1α (*HIF-1α*) via induction of NAD(P)H Oxidase-4 (*NOX4*) and consequent production of ROS. Activation of *HIF-1α* was required for DF-induced

metabolic reprogramming characterized by increased glycolysis and repression of mitochondrial oxidative phosphorylation due to inhibition of pyruvate dehydrogenase (*PDH*) via increased pyruvate dehydrogenase kinase-1 (*PDK1*) expression. Reversal of DF-induced metabolic reprogramming reduced endothelial activation. These new molecular insights identify a previously uncharacterized role of disturbed hemodynamics in stabilizing endothelial *HIF-1α* to dynamically regulate endothelial metabolic plasticity, and consequently, endothelial activation and vascular health.

## Results

### Disturbed flow alters metabolism in human aortic endothelial cells

Human aortic endothelial cells (HAECs) were subjected for 24 hr to 'athero-susceptible' DF mimicking the hemodynamics measured in human carotid sinus or 'athero-protective' UF representing the wall shear stress in human distal internal carotid artery (*Dai et al., 2004*). RNA-seq whole-genome transcriptome profiling and multiple pathway analyses were used to globally determine genes and gene networks that are regulated in endothelium exposed to either DF or UF (*Wu et al., 2017b*). Analysis of RNA sequencing identified a total of 3757 differentially expressed genes (DEGs) using a false discovery rate cut off value of q < 0.05. Gene set enrichment analysis (GSEA) was performed on all DEGs to identify overrepresented biological pathways (*Subramanian et al., 2005*). The DEGs are available at the publisher's website (*Figure 1—source data 1*). Among these pathways, the cellular response to hypoxia and glycolytic metabolism were the most up-regulated gene sets under disturbed flow (*Figure 1A*). The enrichment plot for hypoxia and glycolysis gene sets under DF are provided in *Figure 1—figure supplement 1A and B*, respectively, and list of enriched genes under DF contributing to the hypoxia and glycolytic gene sets are provided in *Figure 1—source data 2*, respectively. As an example, genes traditionally associated with hypoxia (e.g., *VEGFA*) and glycolysis (e.g., *SLC2A1* and *LDHA*) are positively enriched under DF. Functional analysis of the RNA-seq data using DAVID (the Database for Annotation, Visualization, and Integrated Discovery) (*Huang et al., 2009a*), showed that glucose metabolism and response to hypoxia were among the top 10 gene ontology (GO) biological processes induced by DF (*Figure 1B*). In contrast, UF had no effect on these pathways (*Figure 1—figure supplements 2* and *3*). A third pathway analysis was conducted by the bioinformatics tool METASCAPE (*Tripathi et al., 2015*), which also showed that DF induced upregulation of DEGs enriched in glycolytic metabolism (*Figure 1—figure supplement 4*). Hypoxic and glycolytic signature genes are persistent up to 3 days, as shown in *Figure 1—figure supplement 5*.

To determine the metabolic phenotypes of HAECs mediated by athero-relevant flow waveforms, we performed glucose uptake assays and Seahorse bioenergetics measurements (*Pike Winer and Wu, 2014*). Glucose uptake was determined by two approaches: (a) 2-(*N*-(7-Nitrobenz-2-oxa-1,3-diazol-4-yl)Amino)-2-Deoxyglucose (2-NBDG) fluorescence measurement (*Zou et al., 2005*) and (b) 2-deoxyglucose-6-phosphate (2-DG6P), a non-degradable glucose-like substrate which utilizes a colorimetric output (*Han et al., 2015*). Cells subjected to DF exhibited increased glucose uptake as evidenced by increased 2-NBDG fluorescence (*Figure 1C and D*) and 2-DG6P uptake (*Figure 1E*) as compared to UF.

To functionally characterize the DF-induced endothelial metabolic phenotype, HAECs were subjected to a glycolysis stress test or a mitochondrial stress test following 48 hr of exposure to either DF or UF. The glycolysis stress test measures extracellular acidification rate (ECAR, an indicator of glycolytic lactate production) after the addition of glucose (basal glycolysis) and after addition of the mitochondrial inhibitor oligomycin A (glycolytic capacity) (*Pike Winer and Wu, 2014*; *Nigdelioglu et al., 2016*). DF increased both basal glycolysis and glycolytic capacity in HAECs as compared to UF (*Figure 1F,G*). The mitochondrial stress test measures oxygen consumption rate (OCR) at baseline levels, and after addition of oligomycin A (coupled respiration), the mitochondrial uncoupler Carbonyl cyanide-*p*-trifluoromethoxyphenylhydrazone (FCCP) (maximum respiration), and respiratory inhibition with antimycin A and rotenone (non-mitochondrial oxygen consumption) (*Pike Winer and Wu, 2014*). DF reduced basal respiration, maximal respiration, and spare respiratory capacity in HAECs (*Figure 1H,I*). To support these results, we measured the relative dependence of cellular ATP levels on glycolysis and mitochondria as a function of flow type was measured via luminescence (*Borowski et al., 2013*). Whereas under UF mitochondrial oxidative

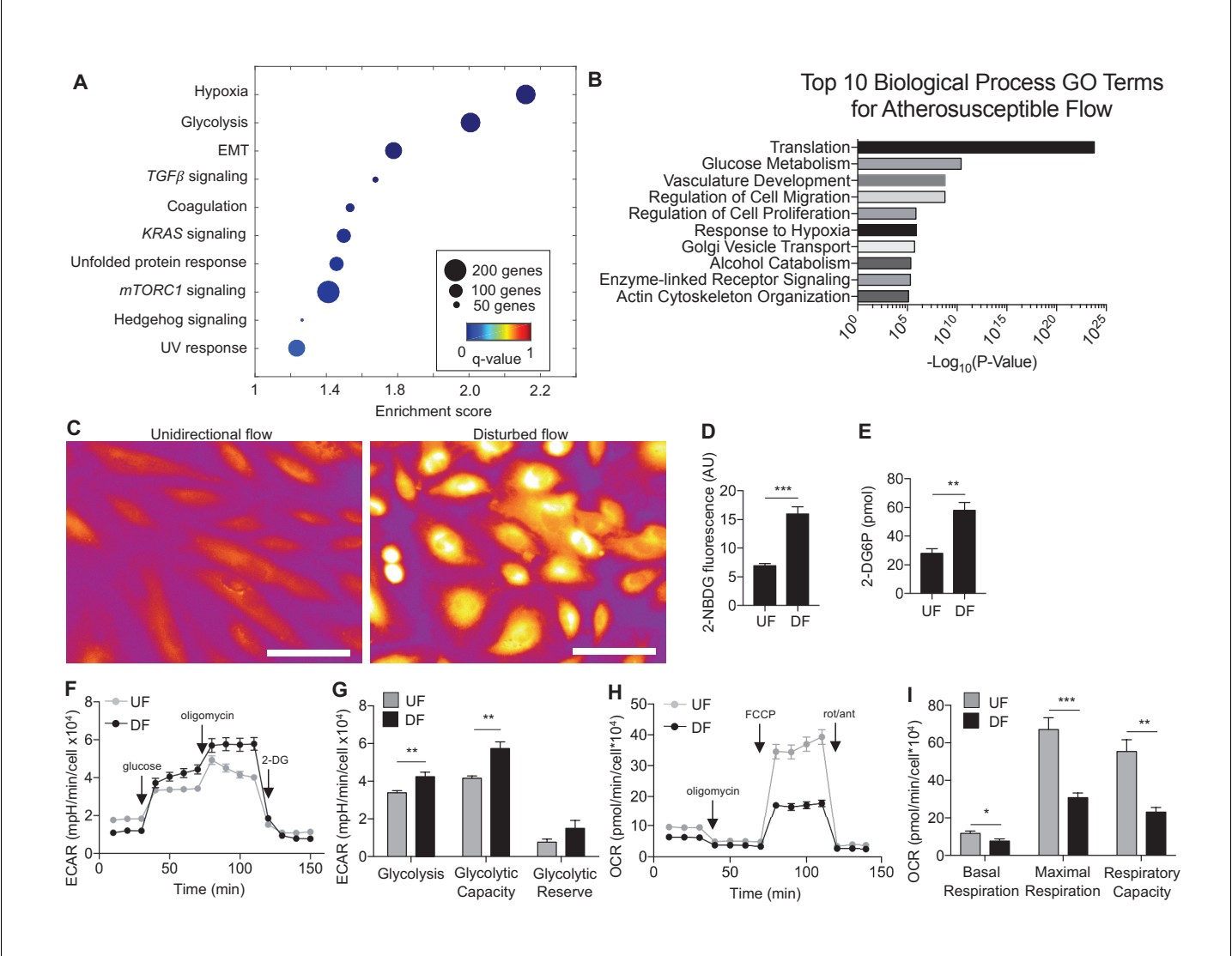

**Figure 1.** Disturbed flow induces a glycolytic phenotype and inhibits mitochondrial function. Primary HAECs were subjected to either unidirectional flow (UF) or disturbed flow (DF) for 24 hr before cell lysates were collected. (**A**) Gene set enrichment analysis of RNA-seq dataset of top 10 pathways enriched under DF compared to UF. (**B**) Top 10 gene ontology pathways of differentially expressed genes as calculated via DAVID. Source data of differentially expressed genes available online. (**C**) HAECs are treated with UF or DF for 48 hr before staining with 2-NBDG. Under DF, the cells are markedly brighter and are quantified **D** (five random fields, 100 cells per condition). (**E**) Intracellular glucose uptake can also be quantified with an uptake assay using 2-DG6P. Under DF, there is increased glucose uptake (n = 4). (**F**) HAECs are treated with either UF or DF for 48 hr before re-plating in a Seahorse XFe24 analyzer and assessed with a glycolysis stress test or (**H**) mitochondrial stress test. In (**F**), arrows indicate injection of glucose, oligomycin A, and 2-deoxyglucose (2DG). (**G**) Under glycolysis stress test, HAECs treated with DF demonstrate increased glycolysis and glycolytic capacity when compared against UF (n = 10). (**H**) Arrows denote addition of oligomycin, carbonyl cyanide-4-(trifluoromethoxy)phenylhydrazone (FCCP), and rotenone/antimycin. (**I**) Under mitochondrial stress test, HAECs treated with DF have decreased basal respiration, maximal respiration, and respiratory capacity when compared against UF (n = 7). *p<0.05; **p<0.005; ***p<0.0005 as determined by Student's t-test. Data represent mean ± SEM. Bar is 10 microns. Source data for RNA-seq differentially expressed genes can be found online: doi: https://doi.org/10.5281/zenodo.260122 (**Wu et al., 2017b**).

The following source data and figure supplements are available for figure 1:

**Source data 1.** Differentially expressed genes after UF or DF and RNAseq.

**Source data 2.** Gene sets for hypoxia and glycolysis as ranked by GSEA of HAECs under disturbed flow.

*Figure 1 continued on next page*

*Figure 1 continued*

**Figure supplement 1.** Gene set enrichment analysis (GSEA) enrichment plots for the hypoxia and glycolysis gene sets in unidirectional vs disturbed flow in HAECs.

**Figure supplement 2.** Gene set enrichment analysis of RNA-seq dataset of top 10 pathways enriched under unidirectional flow compared to disturbed flow.

**Figure supplement 3.** Gene ontology analysis of top 10 biological processes of differentially expressed genes under unidirectional flow using DAVID.

**Figure supplement 4.** Gene ontology analysis of top biological processes of differentially expressed genes under DF by Metascape.

**Figure supplement 5.** Persistence of glycolytic and hypoxia genes under disturbed or unidirectional flow.

**Figure supplement 6.** Mitochondrial ATP production is dependent on flow.

**Figure supplement 7.** Glycolysis stress test and mitochondria stress test for HAEC after 48 hr of static, unidirectional flow, and disturbed flow.

**Figure supplement 8.** Comparison of glycolytic gene expression under disturbed flow, unidirectional flow, and static conditions.

phosphorylation accounted for ~17% of total ATP production, under DF mitochondrial oxidative phosphorylation accounted for less than 1% of the total ATP (*Figure 1—figure supplement 6*). We also performed bioenergetics measurements to compare DF and UF to static, no flow conditions. As shown in *Figure 1—figure supplement 7A and B*, HAECs under static conditions show an intermediate glycolytic and oxidative phenotype. Glycolytic gene expression of static HAECs was also compared against UF and DF at 48 hr in *Figure 1—figure supplement 8*. Static HAECs expressed glycolytic transcriptional responses more similar to UF than DF. Collectively, these results demonstrate that DF increases endothelial glycolysis and reduces mitochondrial oxidative phosphorylation as compared to UF.

## Disturbed flow stabilizes *HIF-1α* in endothelium

Ingenuity Pathway Analysis (IPA) was performed to probe major putative upstream regulators that control the endothelial transcriptomes under DF or UF (*Calvano et al., 2005*). Based on the flow-sensitive 3757 DEGs and *a priori* knowledge of upstream gene regulation, IPA predicted Krüppel-like Factor 2 (*KLF2*) as a major driver of transcriptional changes induced by UF, and *HIF-1α* driving transcriptional changes induced by DF (*Figure 2A*, IPA network genes for top 10 predicted upstream regulators are provided in *Figure 2—source data 1*). *HIF-1α* protein expression was increased in DF-subjected HAECs (*Figure 2B*, *Figure 2—figure supplement 1*). However, DF had no effect on *HIF-1α* mRNA levels (*Figure 2C*) suggesting that DF increases *HIF-1α* protein through a post-transcriptional mechanism. Furthermore, the DF-induced increase in *HIF-1α* protein levels was dynamic and reversible. The DF-induced increase in *HIF-1α* protein could be reversed by switching to UF, whereas the UF-induced reduction could be reversed by switching to DF (*Figure 2D*).

To determine the key biological pathways controlled by *HIF-1α* in endothelium under DF, HAECs were treated with either *HIF-1α* targeting siRNA or control siRNA and then were subjected to DF for 48 hr before performing RNA sequencing (*Wu et al., 2017a*). Transcriptome analyses identified 2989 DEGs regulated by *HIF-1α* in HAECs under DF (FDR < 0.05). GSEA (*Figure 2E*) and DAVID (*Figure 2F*) identified that *HIF-1α*-regulated DEGs are enriched in hypoxia and glycolysis, indicating that *HIF-1α* significantly contributes to the glycolytic gene signature detected in DF-treated HAECs. The DEGs are available at the publisher's website (*Figure 2—source data 2*). The enrichment plot for hypoxia and glycolysis (*Figure 2—figure supplement 2A and B*, respectively), as well as list of enriched genes under disturbed flow and *HIF-1α* knockdown contributing to the hypoxia and glycolytic gene sets are provided (*Figure 2—source data 3*). Genes known to be associated with hypoxia and glycolysis were positively enriched under control siRNA.

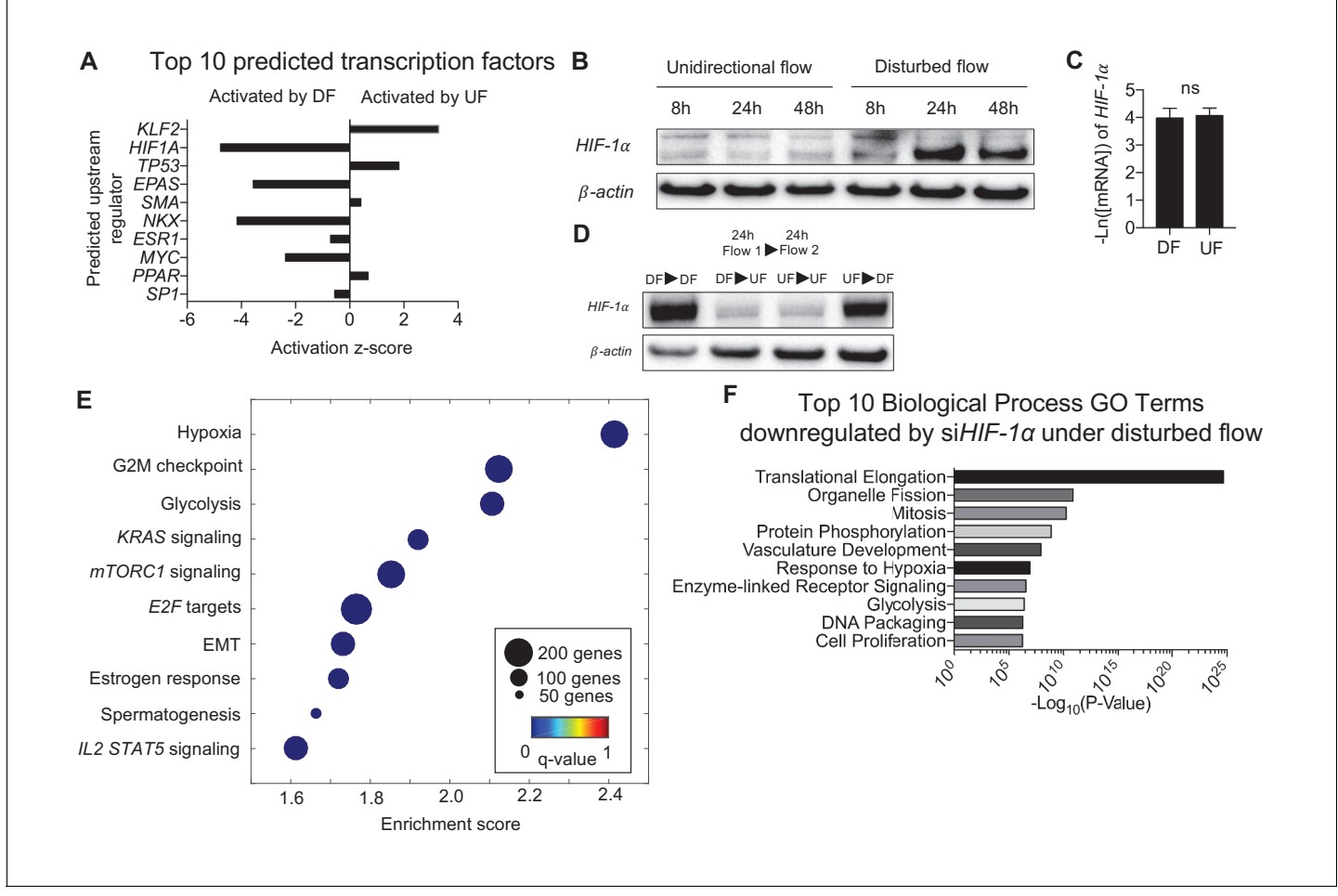

**Figure 2.** Disturbed flow induces *HIF-1α* expression, which accounts for a major portion of the differentially expressed genes under DF. Primary HAECs were subjected to either UF or DF for 24 hr before samples were collected for RNA-seq. (**A**) Using Ingenuity Pathway Analysis, the RNA-seq data was analyzed for transcription factor predictions. The top two predicted transcription factors are *KLF2* (upregulated under UF) and *HIF-1α* (upregulated under DF), as judged by activation z-score. (**B**) Western blot of time course of HAECs subjected to UF or DF. *HIF-1α* starts to appear around 8 hr under disturbed flow. (**C**) qPCR quantification of *HIF-1α* under 48 hr of DF or UF (n = 4). (**D**) HAECs were subjected to either DF or UF for 24 hr, then either UF or DF, respectively, for 24 hr, before lysates were collected. The expression of *HIF-1α* is reversible. (**E**) HAECs were treated with either non-targeting siRNA or si*HIF-1α* for 24 hr before being subjected to AS flow for 48 hr. The cell lysates were then collected and sent for total RNA sequencing. Gene set enrichment analysis of pathways shows that hypoxia and glycolysis are among the top three pathways that are modulated by *HIF-1α* knockdown. (**F**) Gene ontology pathways of the differentially expressed genes downregulated by *HIF-1α* knockdown also demonstrate that metabolic changes via glycolysis is one of the most downregulated pathways. Source data of differentially expressed genes available online. Significance determined by Student's t-test. Data represent mean ± SEM. Source data for RNA-seq differentially expressed genes can be found online: doi: https://doi.org/10.5281/zenodo.260120 (***Wu et al., 2017a***).

The following source data and figure supplements are available for figure 2:

**Source data 1.** Target genes in RNAseq dataset of predicted transcription factors for HAECs treated with UF and DF, by IPA.

**Source data 2.** Differentially expressed genes after DF and si*HIF-1α* and RNAseq.

**Source data 3.** Gene sets for hypoxia and glycolysis as ranked by GSEA of HAECs under disturbed flow and treated with siRNA targeted against HIF-1α.

**Figure supplement 1.** *HIF-1α* under static, unidirectional flow, and disturbed flow.

**Figure supplement 2.** Gene set enrichment analysis (GSEA) for the hypoxia and glycolysis gene sets in disturbed flow with siRNA targeted towards *HIF-1α* or control in HAECs.

# Disturbed flow increases *NOX4* and ROS to stabilize endothelial *HIF-1α*

We then sought to determine the mechanism(s) underlying DF-induced *HIF-1α* stabilization. Generation of reactive oxygen species (ROS) stabilizes *HIF-1α* in a wide range of cells (*Wheaton et al., 2014*; *Weinberg et al., 2010*; *Semenza, 2009*). HAECs exposed to DF had much higher ROS levels as measured by CellRox, a cytoplasmic ROS indicator (*Figure 3A,B*). Treating cells with an antioxidant, EUK134, a synthetic superoxide dismutase/catalase mimetic (*Rong et al., 1999*), reduced *HIF-1α* levels in DF-treated HAECs (*Figure 3C*). Treatment with antioxidant EUK134 had no effect on UF-treated HAECs (*Figure 3—figure supplement 1*).

The RNA-seq data revealed increased expression of *NOX4* in HAECs under DF. As an ROS generating oxidase, *NOX4* is a critical mediator of endothelial inflammation under DF (*Lassègue and Griendling, 2010*; *Hwang et al., 2003*; *Schröder et al., 2012*). Western blot analysis showed that exposure to DF increased the expression of *NOX4* protein (*Figure 3D*). Inhibition of *NOX4* with siRNA reduced DF-induced ROS (*Figure 3E,F*) and *HIF-1α* levels (*Figure 3G*) in HAECs subjected to DF. To further explore the kinetics of *HIF-1α* accumulation and *NOX4* induction, we conditioned

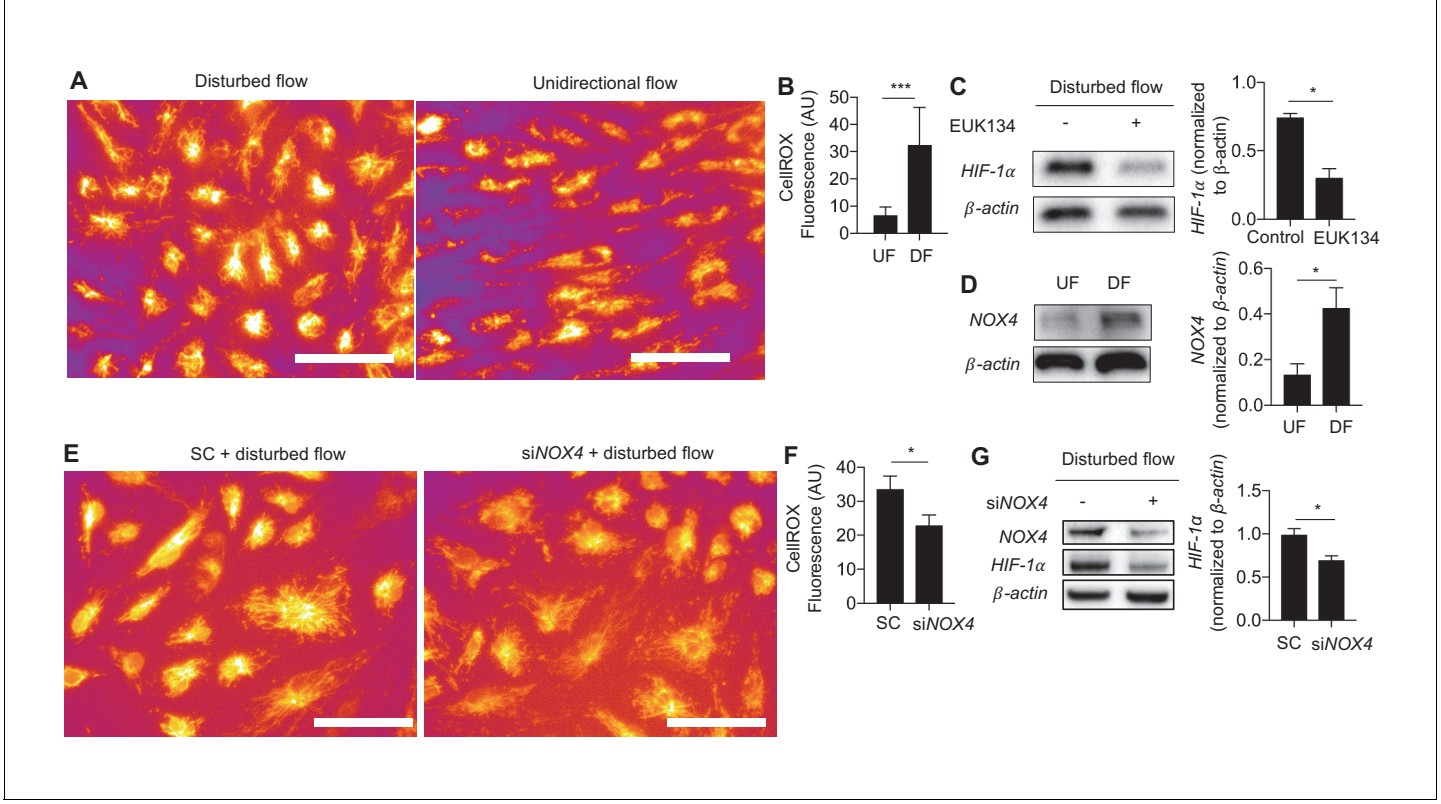

**Figure 3.** Generation of ROS and *NOX4* are required for disturbed flow-induced stabilization of *HIF-1α*. (A) HAECs were subjected to either UF or DF for 48 hr before staining for reactive oxygen species with CellROX orange dye (5 μM). (B) The cells were segmented using the fluorescence channel and the background intensity subtracted before calculating the average fluorescent intensity per cell (n = 50 per condition). (C) HAECs were subjected to DF for 24 hr and simultaneously treated with EUK134 or DMSO. The cells were then lysed and Western blotted for *HIF-1α* and *β-actin*. Treatment with EUK134 under DF reduces *HIF-1α* (n = 4). (D) After 24 hr of either UF or DF, cell lysates were collected for Western blot for *NOX4* (n = 4). HAECs were treated with siRNA targeted towards *NOX4* (si*NOX4*) or non-targeting control (SC) prior to 48 hr of DF followed by either (E) staining for ROS (CellRox, 5 μM), quantified in (F) (n = 50), or cell lysis and Western blotting (G). *NOX4* knockdown reduces *HIF-1α* under DF (n = 4). *p<0.05; **p<0.005; ***p<0.0005 as determined by Student's t-test. Data represent mean ± SEM. Bar is 10 microns.

The following figure supplements are available for figure 3:

**Figure supplement 1.** EUK134 has no effect on HIF-1α under UF.

**Figure supplement 2.** *HIF-1α* and *NOX4* kinetics under DF.

HAECs with 24 hr of UF prior to decreasing durations of DF and measured the expression of these two proteins. As shown in *Figure 3—figure supplement 2A*, *HIF-1α* is maximally induced at 4 hr before stabilizing at the levels seen at 24 hr. *NOX4*, on the other hand, steadily accumulates over time. In order to answer whether *NOX4* levels are upstream of *HIF-1α* at these early time points, or if early *HIF-1α* induction is independent of *NOX4*, we knocked down *NOX4* with siRNA for 24 hr before subjecting these HAECs to decreasing amounts of DF. As can be seen in *Figure 3—figure supplement 2B*, even at early time points, prior to maximal amounts of *NOX4*, knocking down *NOX4* reduces *HIF-1α*. Thus, *NOX4* activation by DF is required for initial induction of *HIF-1α*.

## Disturbed flow stimulates glycolysis by stabilizing *HIF-1α*

The combinatorial analyses of the two above mentioned RNA-seq experiments identified a cohort of glycolytic activators that are increased by DF and induced by *HIF-1α*. Transcripts of 18 glycolytic enzymes were significantly increased (FDR < 0.05) in HAECs under DF when compared to UF and 13 of these enzymes were significantly reduced when DF induced *HIF-1α* was inhibited by siRNAs. Shown in *Figure 4A and B* are the biological and technical replicates for the mRNA fold changes for these genes. Data for combined biological and technical replicates is available in *Figure 2—source data 2*. Among these DF-activated, *HIF-1α*-dependent genes, we found that hexokinase-2 (*HK2*), and glucose transporter-1 (*SLC2A1/GLUT1*) were the most differentially expressed at the transcription level by DF. Confirming the RNA-seq data, we found that disturbed flow increased the mRNA and protein expression of endothelial *SLC2A1* and *HK2* (*Figure 4C,D*). Transfecting HAECs with *HIF-1α*-targeting siRNAs reduced or prevented DF-induced *SLC2A1* and *HK2* mRNA and protein expressions (*Figure 4E,F*). We then determined whether *HIF-1α* expression is sufficient to induce *SLC2A1* and *HK2*. To this end, we mutated the specific proline residues that regulate *HIF-1α* stability (*Ke and Costa, 2006*) to generate a stabilized *HIF-1α* mutant (*mHIF-1α*). *mHIF-1α* transfected HAECs showed high expression of *HIF-1α* (*Figure 4—figure supplement 1*). Compared to control, *mHIF-1α* transfected HAECs had higher mRNA expression of *SLC2A1* and *HK2* (*Figure 4G*).

To determine whether *HIF-1α* is required for DF-induced increase in glycolysis, we performed a glycolysis stress test in HAECs under DF in combination with inhibition of *HIF-1α* by siRNA. *HIF-1α* knockdown significantly reduced the basal glycolysis and glycolytic capacity in HAECs subjected to DF for 48 hr (*Figure 3I,J*). To assess whether *EPAS1* (*HIF-2α*) contributes to the glycolytic phenotype under DF, as *EPAS1* is also predicted to be a transcriptional regulator from IPA (*Figure 2A*), we first verified that *EPAS1* was increased under DF (*Figure 4—figure supplement 2A*), before treating HAECs with siRNA targeted towards *EPAS1* (si*EPAS1*) (*Figure 4—figure supplement 2B*); however, there was no statistically significant reduction in glycolytic genes *HK2* and *SLC2A1* (*Figure 4—figure supplement 2C*) as the result of *EPAS1* inhibition in cells under DF. Together, these results demonstrate that *HIF-1α* but not *EPAS1* (*HIF-2α*) is required for the DF-induced increase in two of the most highly transcriptionally regulated glycolytic enzymes.

## Disturbed flow inhibits mitochondrial respiration via *HIF-1α*-induced *PDK1* activation

We next sought to determine the contribution of *HIF-1α* in DF-induced reduction in mitochondrial respiration (*Figure 1H,I*). We performed a mitochondrial stress test in HAECs treated with control or *HIF-1α*-targeting siRNAs and then were subjected to DF for 48 hr. *HIF-1α* knockdown reversed the DF-induced reduction in maximal respiratory capacity (*Figure 5A*), and increased the basal oxygen consumption and reserve respiratory capacity (*Figure 5B*). The amount of maximal respiration recovered via *HIF-1α* knockdown is substantial, suggesting that *HIF-1α* reduces respiratory capacity in HAECs under DF.

We then determined whether changes in mitochondrial mass contribute to the reduced endothelial respiratory capacity under DF. In HAECs subjected to either DF or UF for 48 hr, no difference was noted in mitochondria density by MitoTracker staining, a surrogate for mitochondria mass (*Figure 5—figure supplement 1A*) (*Agnello et al., 2008*). Furthermore, there was also no difference in transcript levels of mitochondrially encoded genes such as NADH-ubiquinone oxidoreductase chain 1 (*ND-1*) and 5 (*ND-5*) under athero-relevant flows (*Figure 5—figure supplement 1B*).

Substrate limitations or substrate import into mitochondria could account for the decreased mitochondrial respiration in HAEC under DF. To delineate these possibilities, oxygen consumption rates

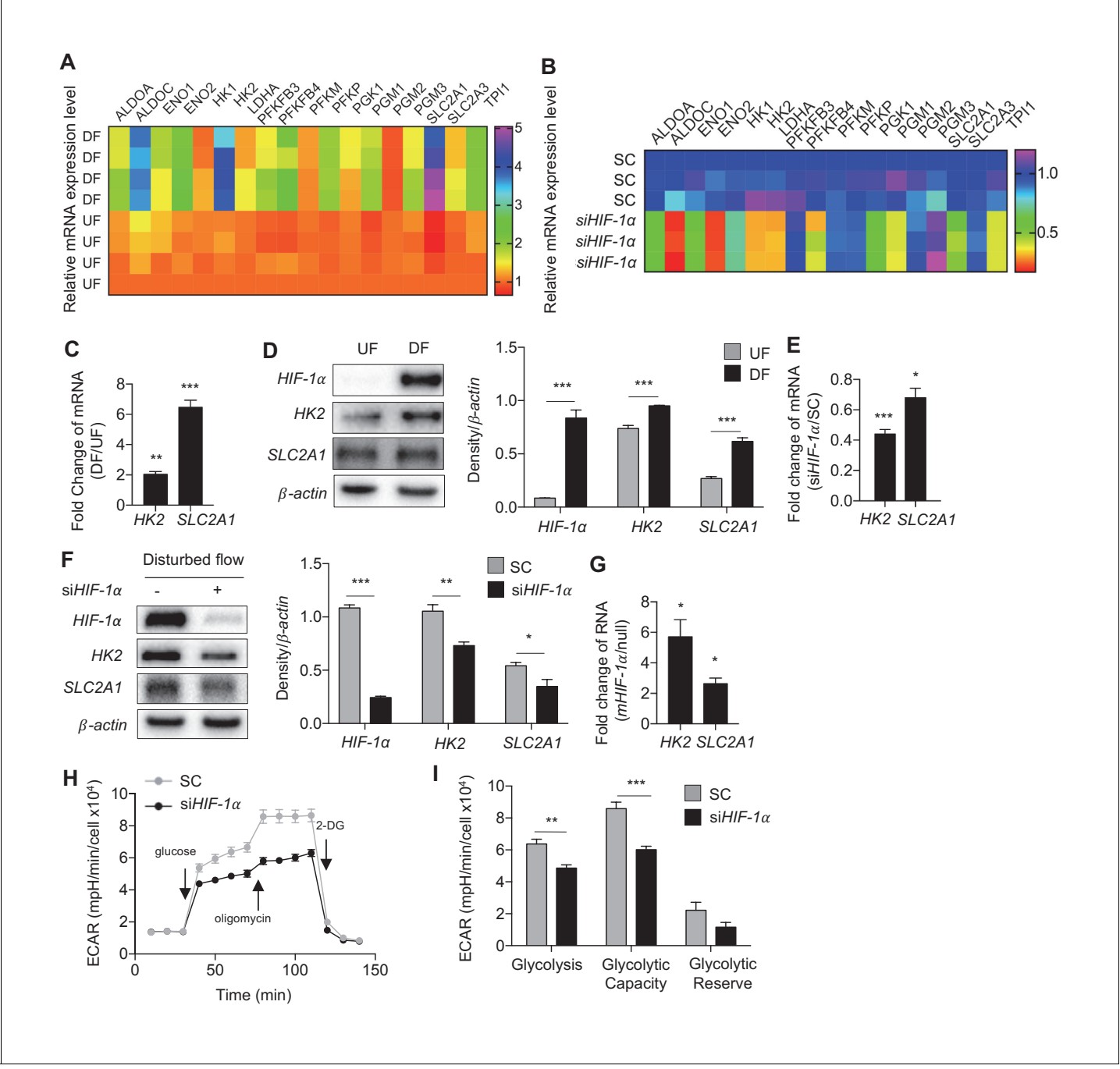

**Figure 4.** Disturbed flow-induced *HIF-1α* stabilization is required for the glycolytic phenotype. (**A**) Expression profile of all significantly regulated glycolytic enzymes in the RNAseq data set of UF vs DF (flow-seq). Four biological replicates for each condition, and three averaged technical replicates. *SLC2A1* and *HK2* are the top two enzymes that are upregulated under DF. Relative expression is normalized to UF, last row. HAECs were treated with siRNA for *HIF-1α* (si*HIF-1α*) or non-targeting control (SC) for 24 hr before DF for an additional 48 hr. Cell lysates were then collected and sent for total RNA sequencing (si*HIF-1α*/DF-seq). (**B**) Expression profile of all significantly regulated glycolytic enzymes in si*HIF-1α*/DF-seq. Three biological replicates for each condition, and two averaged technical replicates. Relative expression is normalized to SC, first row. (**C**) *SLC2A1* and *HK2* upregulated in flow-seq were confirmed with qPCR (n = 4). (**D**) Western blot and quantification of select glycolytic enzymes (*SLC2A1* and *HK2*) under differential flow after 24 hr (n = 4). HAECs were treated with si*HIF-1α* or SC for 24 hr before DF for an additional 48 hr before cell lysis and (**E**) qPCR for *HK2* and *SLC2A1* (n = 13) or (**F**) Western blotting/quantification of *HIF-1α*, *HK2*, *SLC2A1* and *β-actin* (n = 4). (**G**) An overexpression vector containing *HIF-1α* with mutated known prolyl hydroxylase binding amino acids was expressed in HAECs using in vitro transcription for 6 hr before cell lysis and analysis of *HIF-1α*, *SLC2A1* (*β-actin* serves as a loading control) and qRT-PCR analysis for glycolytic genes *SLC2A1* and *HK2* at 50 ng dose (n = 3). (**H**) HAECs were first treated with either SC or si*HIF-1α* before being subjected to DF for 48 hr. The cells then underwent a glycolysis stress test. Arrows indicate injection of

*Figure 4 continued on next page*

*Figure 4 continued*

glucose, oligomycin, and 2-deoxyglucose (2DG). (I) Glycolysis and glycolytic capacity obtained during glycolysis stress test are significantly downregulated by *HIF-1α* knockdown (n = 10). *p<0.05; **p<0.005; ***p<0.0005 as determined by Student's t-test. Data represent mean ± SEM.

The following figure supplements are available for figure 4:

**Figure supplement 1.** Disturbed flow stabilization of *HIF-1α* precedes NOX4 abundance.

**Figure supplement 2.** Disturbed flow stabilization of EPAS1 does not contribute to glycolytic gene transcription.

(OCR) were monitored in HAECs while inhibiting pyruvate entry into mitochondria with UK5099, which inhibits the mitochondrial pyruvate carrier, glutamine entry into mitochondria via BPTES, and fatty acid entry into mitochondria via etoxomir. As demonstrated in *Figure 5C*, glucose oxidation (as estimated by UK5099 treatment) contributes to 43% of the total oxygen consumption rate under UF but confers 62% of total OCR under DF. These findings suggest that decreased pyruvate availability or pyruvate dehydrogenase activity might therefore limit the maximal OCR under UF.

We then sought to determine how DF affects pyruvate availability and pyruvate dehydrogenase (*PDH*), which converts pyruvate into acetyl-CoA. Activity of *PDH* is tightly regulated by pyruvate dehydrogenase kinase-1 (*PDK1*), which inhibits *PDH*. Among the 4 *PDK* genes (*PDK1-4*) expressed in HAECs, *PDK1* was the only one that is upregulated by DF in the RNA-seq data. Consistent with these data, DF increased *PDK1* mRNA (*Figure 5—figure supplement 2*) and protein expression (*Figure 5D*). Loss of *HIF-1α* in HAECs attenuated DF-induced upregulation of *PDK1* mRNA (*Figure 5E*) and protein (*Figure 5F*).

To determine whether the DF-induced increase in *PDK1* affects *PDH* activity, we measured *PDH* activity (*Hong et al., 2013*). Consistent with the increased *PDK1* expression, DF increased *PDH* activity compared to UF (*Figure 5G*). Studies in HAECs with *HIF-1α* knockdown and *PDK1* knockdown showed partial (*Figure 5H*) and complete reversal (*Figure 5I*) of the DF-induced reduction of *PDH* activity, respectively. To test the putative role of *PDK1* in inhibiting endothelial mitochondrial respiration, we subjected HAECs to DF in the presence or absence of dichloroacetate, a known inhibitor of *PDK1* (*Xie et al., 2011*), and performed a mitochondrial stress test. *PDK1* inhibition partially rescued the reduced maximal respiration and respiratory capacity induced by DF (*Figure 5J–K*). Similar results were obtained with genetic inhibition of *PDK1* expression by siRNA (*Figure 5L–M*, *Figure 5—figure supplement 3*). Collectively, these findings suggest that under DF, *HIF-1α* promotes *PDK1* expression to reduce the entry of glucose-derived pyruvate into the TCA cycle, and consequently limits mitochondrial reserve function.

## Disturbed flow-induced metabolic reprogramming is required for endothelial activation

To determine how DF-induced changes in cellular metabolism affect inflammatory gene expression in endothelial cells, we measured the expression of pro-inflammatory cytokines in DF-activated HAECs. Exposure of HAECs to DF for 48 hr resulted in increased expression of pro-inflammatory genes including *VCAM-1*, *IL8* and *CCL2* compared to UF (*Figure 6A*). Inhibition of *HIF-1α* by siRNA reduced DF-induced expression of *IL8*, *VCAM-1*, and *CCL2* (*Figure 6B*). Knocking down *HIF-1α* under disturbed flow also reduced the binding of monocytes (THP-1) to the HAECs (*Figure 6—figure supplement 1*). Conversely, ectopic expression of stabilized *HIF-1α*, like DF, increased *IL8*, *VCAM-1*, and *CCL2* expression in HAECs (*Figure 6C*). Moreover, siRNA-mediated *NOX4* inhibition, which prevented the DF-induced *HIF-1α*, reduced the expression of pro-inflammatory genes in DF-treated HAECs (*Figure 6D*).

We then investigated whether modulation of metabolism influences endothelial inflammatory gene expression. To inhibit DF-induced glycolysis, we treated HAECs with siRNA against *SLC2A1* (si*SLC2A1*), which was the most highly upregulated glycolytic gene under DF. *SLC2A1* knockdown cells had reduced *SLC2A1* mRNA and protein levels, and exhibited reduced ECAR when compared with control knockdown cells (*Figure 6—figure supplement 2*). Inhibition of *SLC2A1* (or glycolysis) reduced DF-induced *VCAM-1* and *CCL2* gene expression (*Figure 6E*). To rescue the DF-induced

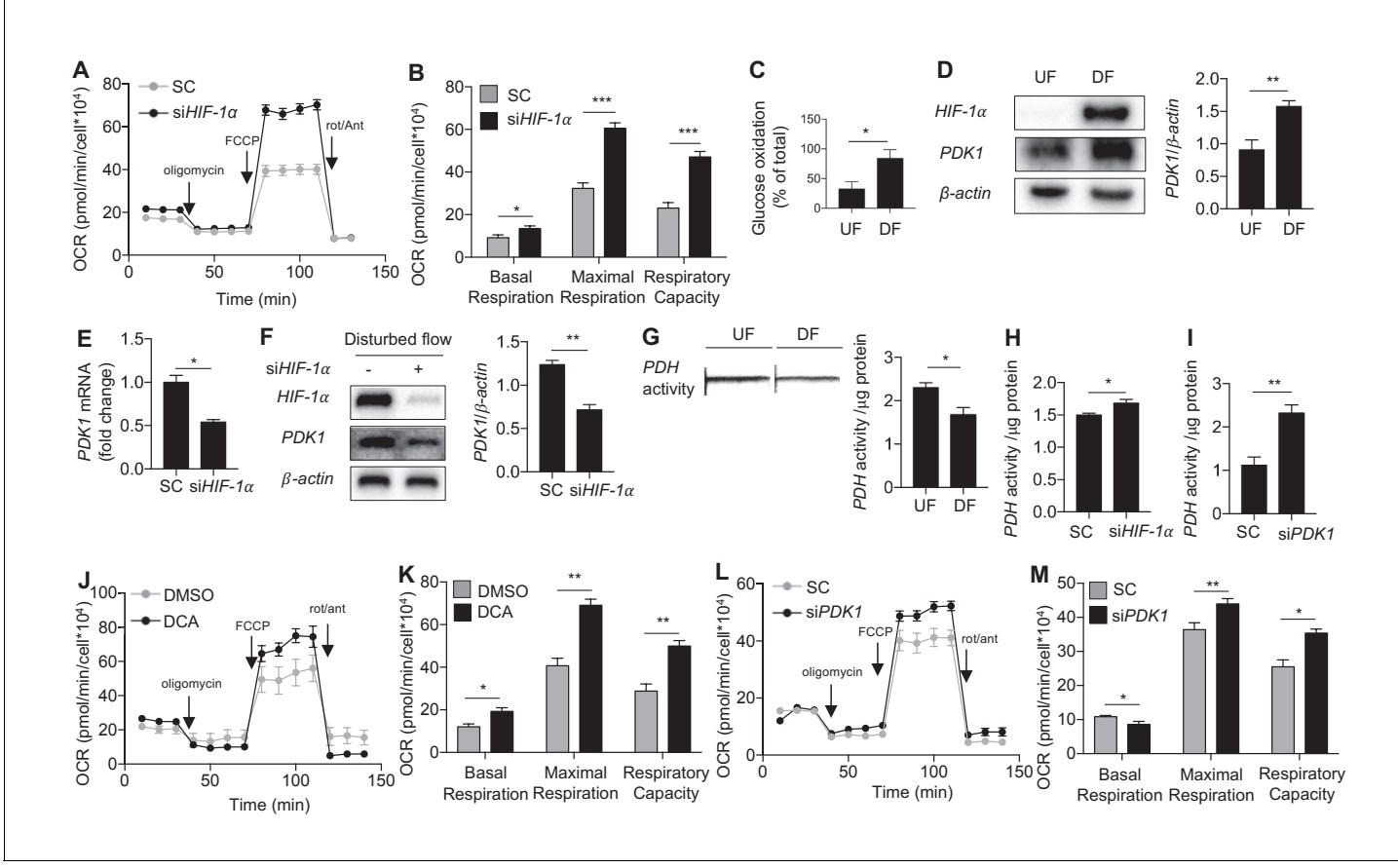

**Figure 5.** Disturbed flow inhibits mitochondrial respiration via HIF-1α. (**A**) HAECs were subjected to either DF for 48 hr following 24 hr treatment with either non-targeting siRNA (SC) or *HIF-1a* targeted siRNA (si*HIF-1α*), followed by a mitochondrial stress test. (**B**) Respiratory parameters (**A**) are all significantly higher under DF cells treated with si*HIF-1α* (n = 8). (**C**) HAECs were subjected to either UF or DF for 48 hr. Glucose oxidation was measured during sequential treatment with UK5099 and BPTES/Etoxomir. Under DF, HAECs use glucose as a larger fraction of the total mitochondrial oxidation (n = 4). HAECs were subjected either 24 hr UF or DF. Western blot (**D**) and qRT-PCR (**E**) shows increased *PDK1* expression under DF (n = 4 for both). (**F**) HAECs were subjected to DF for 48 hr following 24 hr treatment with either SC or si*HIF-1α*. Cell lysates demonstrate reduced *PDK1* under si*HIF-1α* (n = 4). (**G**) HAECs were subjected to 48 hr of either UF or DF before undergoing a *PDH* activity dipstick assay. DF reduces *PDH* activity (n = 4). (**H**) Under combined si*HIF-1α* treatment and DF, there is increased *PDH* activity (n = 4). (**I**) Under combined si*PDK1* treatment and DF, there is increased *PDH* activity (n = 4). (**J**) HAECs are treated with either DMSO or DCA (4 mM) and simultaneous DF for 48 hr before mitochondrial stress test. Respiratory parameters are shown in **K**. There is a signficant increase in maximal respiration and respiratory capacity with DCA treatment (n = 4). (**L**) HAECs were treated with SC or si*PDK1* for 24 hr before DF for 48 hr and subsequent mitochondrial stress test. Respiratory parameters are shown in **M** (n = 8). *p<0.05; **p<0.005; ***p<0.0005 as determined by Student's t-test. Data represent mean ± SEM.

The following figure supplements are available for figure 5:

**Figure supplement 1.** Mitochondrial biogenesis is the same in differential flow.

**Figure supplement 2.** Differential flow regulates *PDK1*.

**Figure supplement 3.** siRNA targeted against *PDK1*.

reduction in mitochondrial oxidative phosphorylation, we treated HAECs with siRNA against *PDK1*. Inhibition of *PDK1* also significantly reduced inflammatory gene expression (*Figure 6F*). To further explore whether the inhibition of oxidative phosphorylation by itself would further increase inflammation, we treated HAECs with a combination of rotenone and antimycin which together halt electron flow in the mitochondria (*Chen et al., 2003*). We found that inhibition of the electron transport chain with rotenone and antimycin increased *IL8*, *VCAM1*, and *CCL2* in UF (*Figure 6—figure*

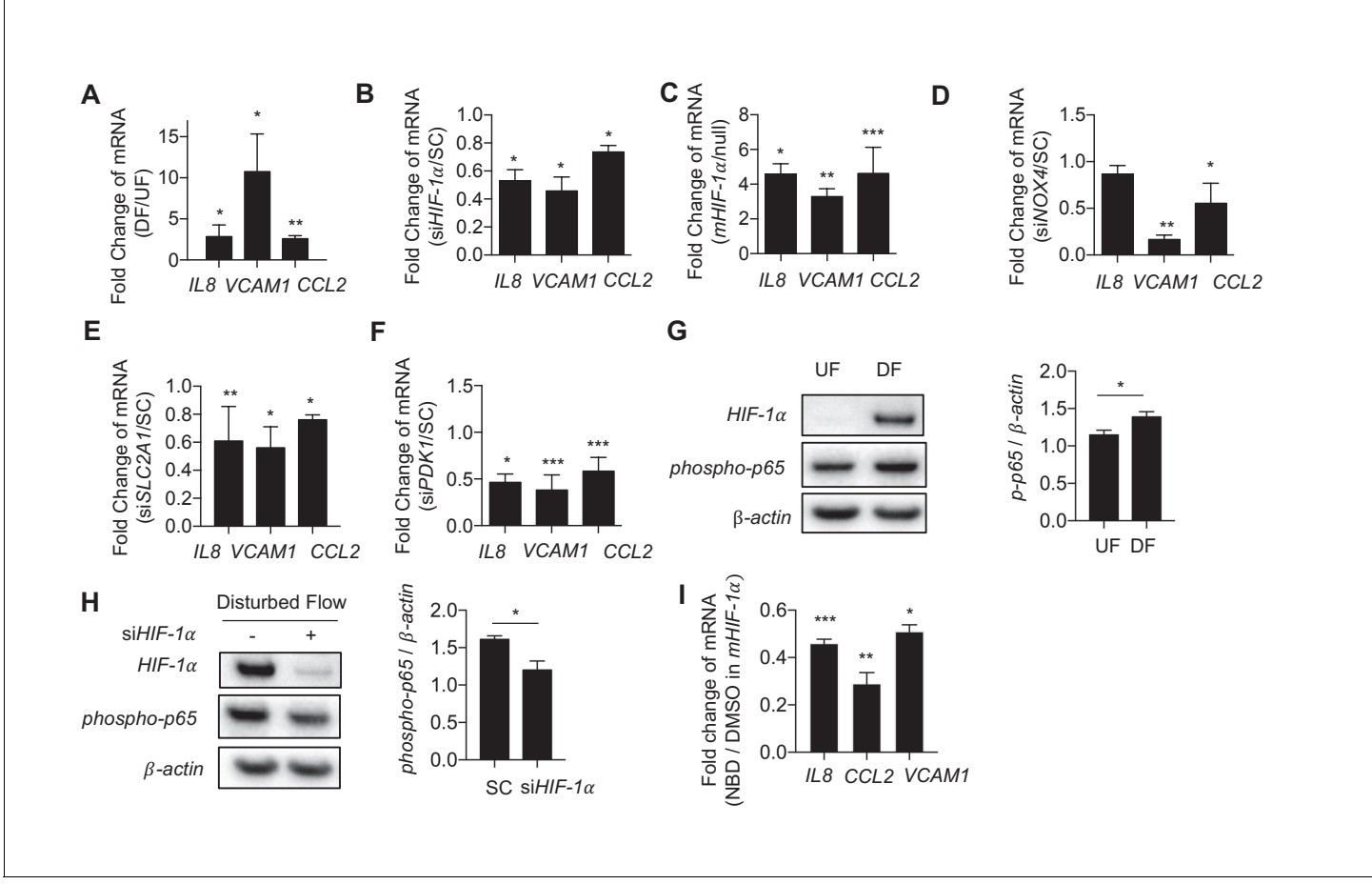

**Figure 6.** Disturbed flow-induced metabolic reprogramming is required for expression of inflammatory markers. (**A**) HAECs were treated with either UF or DF for 24 hr prior to lysis, RNA purification, and qRT-PCR for *IL8*, *VCAM1*, and *CCL2*. *IL8*, *VCAM1*, and *CCL2* are all increased under DF (n = 4). (**B**) HAECs were treated with either non-targeting siRNA (SC) or siRNA targeting *HIF-1α* (si*HIF-1α*) for 24 hr before subjected to DF for 48 hr prior to lysis, RNA purification, and qRT-PCR for *IL8*, *VCAM1*, and *CCL2*. *IL8*, *VCAM1*, and *CCL2* are all reduced under *HIF-1α* knockdown (n = 4). (**C**) HAECs were transfected with either a blank or with stabilized *HIF-1α* mRNA transcript for 6 hr prior to lysis, RNA purification, and qRT-PCR for *IL8*, *VCAM1*, and *CCL2*. *IL8*, *VCAM1*, and *CCL2* are all increased under *HIF-1α* transcription delivery (n = 4). (**D**) HAECs were treated with either non-targeting siRNA (SC) or siRNA targeting *NOX4* (si*NOX4*) for 24 hr before subjected to DF for 48 hr prior to lysis, RNA purification, and qRT-PCR for *IL8*, *VCAM1*, and *CCL2*. *IL8*, *VCAM1*, and *CCL2* are all reduced under *NOX4* knockdown (n = 4). (**E**) HAECs were treated with either non-targeting siRNA (SC) or siRNA targeting *SLC2A1* (si*SLC2A1*) for 24 hr before subjected to DF for 48 hr prior to lysis, RNA purification, and qRT-PCR for *IL8*, *VCAM1*, and *CCL2*. *IL8*, *VCAM1*, and *CCL2* are all reduced under *SLC2A1* knockdown (n = 4). (**F**) HAECs were treated with either non-targeting siRNA (SC) or siRNA targeting *PDK1* (si*PDK1*) for 24 hr before subjected to DF for 48 hr prior to lysis, RNA purification, and qRT-PCR for *IL8*, *VCAM1*, and *CCL2*. *IL8*, *VCAM1*, and *CCL2* are all reduced under *PDK1* knockdown. (**G**) HAECs were treated with either UF or DF for 24 hr prior to lysis and Western Blotting (n = 4). (**H**) si*HIF-1α* treated HAECs were subjected to DF for 48 hr prior to lysis. si*HIF-1α* significantly reduces *phospho-p65* (n = 4). (**I**) HAECs were treated with 10 μM of NBD for 1 hr prior to *HIF-1α* overexpression for 6 hr prior to RNA purification and qRT-PCR (n = 4). *p<0.05; **p<0.005; ***p<0.0005 as determined by Student's t-test. Data represent mean ± SEM.

The following figure supplements are available for figure 6:

**Figure supplement 1.** *HIF-1α* knockdown reduces leukocyte adhesion.

**Figure supplement 2.** *SLC2A1* controls glycolysis in HAECs.

**Figure supplement 3.** Mitochondrial inhibition increases inflammation.

**Figure supplement 4.** *HIF-1α* is upstream of *NF-κB*-induced inflammatory gene transcription.

**Figure supplement 5.** Disturbed flow stabilization of *HIF-1α* reduces *KLF2*.

*supplement 3*). These results are consistent with si*HIF-1α* and si*PDK1* improving mitochondrial capacity, as well as reducing inflammation, under DF. In summary, these findings suggest that DF-induced metabolic reprogramming; namely induction of glycolysis and inhibition of mitochondrial oxidation, is required for the DF-induced pro-inflammatory phenotype.

*NF-κB* has been reported to be an important regulator of endothelial inflammation under DF (*Hajra et al., 2000*). Indeed, we found that DF causes increased phosphorylation of the p65 subunit of *NF-κB* (*Figure 6G*). Knock down *HIF-1α* reduced DF-induced *NF-κB* activation (*Figure 6H*). To further confirm the role of *HIF-1α* in *NF-κB* activation, we treated HAECs with NEMO-binding domain peptide (NBD), which blocks association of NEMO with *IκB* kinase that phosphorylates *IκB*, thereby preventing *NF-κB* activation (*May et al., 2002*). We then over-expressed *HIF-1α* (*mHIF-1α*) in these NBD-treated HAECs. Pretreatment of NBD prevented the *HIF-1α*-induced increase in inflammation (*Figure 6I*). We next examined whether *NF-κB* activation was required for *HIF-1α* stabilization, as has been suggested by others (*van Uden et al., 2008*, *2011*). Pretreatment of HAECs with NBD prior to 24 hr of DF did not affect DF-induced increase in *HIF-1α* levels, suggesting that *NF-κB* activation is not responsible for *HIF-1α* stabilization in HAECs under DF (*Figure 6—figure supplement 4*).

DF is also associated with decreased *KLF2* expression compared to UF, and *KLF2* is a known inhibitor of NF-κB activation (*SenBanerjee et al., 2004*). In our RNAseq dataset, *KLF2* was downregulated 11-fold under DF when compared to UF (FDR < 0.000345). Furthermore, *KLF2* is known to inhibit *HIF-1α* under hypoxic conditions (*Kawanami et al., 2009*). We thus asked whether *HIF-1α* induction decreases *KLF2* levels under DF. As shown in *Figure 6—figure supplement 5A*, knockdown of *HIF-1α* under DF increased *KLF2* levels (but not *KLF4*) (*Figure 6—figure supplement 5B*). Of note, in the RNAseq DEG dataset of si*HIF-1α* under DF, both *KLF2* and *KLF4* were significantly upregulated, compared to scrambled control. Together, these results suggest that DF-induced metabolic changes driven by *HIF-1α* cause increased inflammation through an *NF-κB*-dependent mechanism.

## Expression of *NOX4*, *ROS*, *HIF-1α*, *phospho-p65*, and glycolytic enzyme expression are increased in the inner curvature of the aortic arch where blood flow is disturbed

To investigate in vivo relevance of DF-mediated regulation of endothelial metabolism, we measured the expression of glycolytic enzymes and ROS in endothelial cells freshly isolated from different regions of porcine aorta of known susceptibilities to atherosclerosis. The inner curvature of the aortic arch is constantly exposed to DF, and prone to atherosclerosis while the nearby descending thoracic aorta is subjected to UF, and resistant to atherosclerosis (*Figure 7A*). *En face* images showed that endothelia located in aortic arch exhibited cobblestone morphology and greater amounts of ROS (measured by dihydroethidium bromide) (*Figure 7B*) compared to elongated endothelia in the nearby descending aorta (*Figure 7C*). Endothelial cells (95% purity) were scraped immediately after the swine were sacrificed for Western Blot analyses (*Wu et al., 2015*; *Fang et al., 2010*). As shown in *Figure 7D and E*, we detected significantly increased protein expression of *NOX4*, *HIF-1α* and glycolytic enzymes *SLC2A1*, *HK2*, and *PDK1*, as well as *phosho-p65*, in endothelia isolated form the inner curvature of swine aortic arch as compared to those isolated from descending thoracic aorta in the same animal. Similarly, using immunofluorescence, we confirmed that *NOX4* (*Figure 7—figure supplement 1A,B*) and *HIF-1α* were increased in parts of murine aortas, after live perfusion with fixative followed by immunofluorescence (*Figure 7—figure supplement 1C,D*). These in vivo results are consistent with the in vitro studies demonstrating that disturbed flow promotes endothelial glycolytic metabolism and reduced mitochondrial function by increasing *NOX4*-mediated ROS production and consequent *HIF-1α* stabilization.

## Discussion

We discovered a previously uncharacterized role for *HIF-1α* in metabolic reprogramming and activation of vascular endothelium under disturbed flow, which simulates hemodynamics associated with atherosclerosis. Although flow-dependent metabolic plasticity has been implicated in endothelial pathophysiology (*Doddaballapur et al., 2015*; *Cucullo et al., 2011*), the underlying molecular mechanisms are poorly understood. Here, we evaluated the DF-induced regulation of endothelial

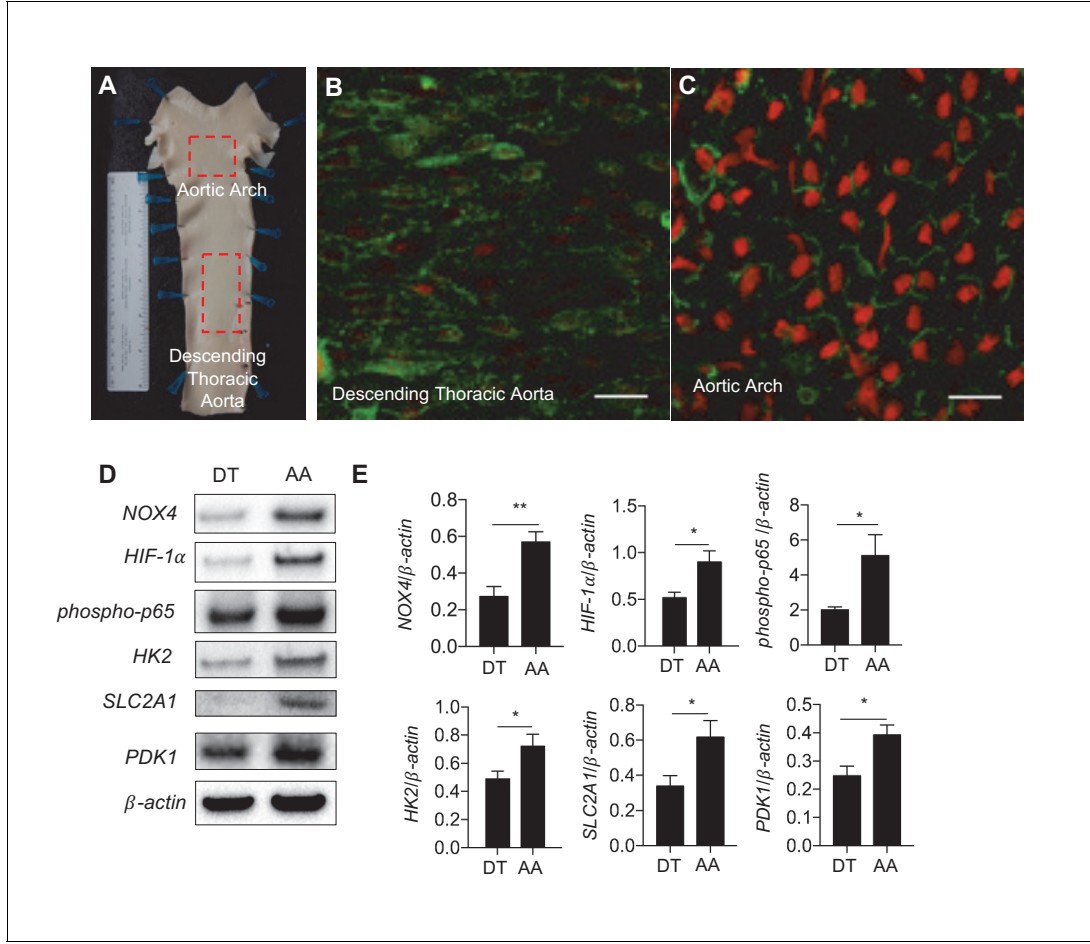

**Figure 7.** Athero-susceptible regions of aorta express higher levels of ROS and HIF-1α. (**A**) Aortas were harvested from pigs due for slaughter in less than 10 min after sacrifice. The inner curvature (**B**) of the aortic arch (AA) or descending thoracic (DT) aorta (**C**) was cut out and immediately stained with dihydroethidium bromide (DHE) before fixation in cold 4% paraformaldehyde. The sections were then permeabilized and stained with lectin. Other aortas were dissected out and immediately washed with PBS before a #10 scalped was passed along the inner curve of the aorta or along the descending thoracic aorta. Endothelial cells were immediately stored in cold lysis buffer. (**D**) Western blotting for *NOX4, HIF-1α, phospho-p65, HK2, SLC2A1, PDK1,* and *β-actin* of AA and DT samples. (**E**) The Western blots are quantified for *NOX4, HIF-1α, phospho-p65, HK2, SLC2A1,* and *PDK1* (n = 4). The AA region of the pig aortas have significantly more expression of all these enzymes. *p<0.05; **p<0.005 as determined by Student's t-test. Data represent mean ± SEM. Bar is 10 microns.

The following figure supplement is available for figure 7:

**Figure supplement 1.** *NOX4* and *HIF-1α* immunofluorescence in descending thoracic aorta (DT) and aortic arch (AA).

metabolism by a combination of whole-genome RNA sequencing, pathway predictions, bioenergetics measurements, and molecular/biochemical analyses. We found that DF increases endothelial glycolysis and reduces mitochondrial respiratory capacity by promoting *NOX4*-dependent ROS-mediated *HIF-1α* stabilization. In agreement with the in vitro investigations, *NOX4*, ROS, *HIF-1α*, and key glycolytic activators were also increased in vivo in endothelial cells isolated from the inner curvatures of porcine aortic arch, which represents an area where cells are exposed to DF when compared to nearby descending thoracic aorta where cells are subjected to UF. These findings demonstrate that a metabolic switch comprising *HIF-1α*-dependent reduction in respiratory capacity and increased glycolysis as important molecular signatures that characterize the athero-susceptible endothelium associated with DF.

Our RNA-seq analyses predicted that *KLF2* and *HIF-1α* are the major transcriptional regulators that control the endothelial phenotypes related to athero-relevant flows. Subsequent mechanistic

investigations further demonstrate that DF-induced expression of *HIF-1α* promotes endothelial glycolytic metabolism and reduces mitochondrial oxidative phosphorylation. Our data are consistent with reports on glycolysis playing a central role in the promotion of angiogenesis (*De Bock et al., 2013b*; *Potente et al., 2011*). However, these previous studies did not apply differential flow, which has long been known to result in different EC phenotypes. Our data are also consistent with reports of mitochondria acting as signaling organelles (*Quintero et al., 2006*); the acute reduction in mitochondrial oxidation under DF may act as a shunt to oxygenate tissues further away from the vasculature.

Hypoxia-inducible factors (*HIF*s) function as master regulators of oxygen homeostasis, and *HIF-1α* and *HIF-2α* are the most widely studied members of the *HIF* family (*Semenza, 2012*; *Prabhakar and Semenza, 2012*). While *HIF-1α* is expressed in all cells of all metazoan species, *HIF-2α* is only expressed in certain cell types of vertebrate species including endothelial cells. Unlike *HIF-2α*, *HIF-1α* controls the expression of multiple gene products that mediate glycolysis (*Prabhakar and Semenza, 2012*). For instance, *HIF-1α* activates transcription of *PDK1*, which phosphorylates and inactivates the catalytic subunit of *PDH*, the enzyme that converts glucose-derived pyruvate into acetyl-CoA for entry into the mitochondrial tricarboxylic acid (TCA) cycle (*Kim et al., 2006*). Inhibition of *PDK1* only partially restored the mitochondrial reserve capacity in HAECs under DF compared to *HIF-1α* knockdown, which almost completely rescued mitochondrial reserve capacity to the level detected under UF. These findings indicate that besides *PDK1*, *HIF-1α* exerts its effects on mitochondrial oxidative phosphorylation via additional signaling pathways. One possibility is the flow-mediated regulation of Lactate Dehydrogenase A (*LDHA*), a key enzyme that catalyzes the conversion of pyruvate to lactate in the final step of anaerobic glycolysis. Indeed, our RNA-seq analyses demonstrated up-regulation of *LDHA* expression by DF; this effect was suppressed by *HIF-1α* inhibition. Increased *LDHA* is predicted to reduce the availability of pyruvate to enter the TCA cycle and hence limit the substrate availability for oxidative phosphorylation.

Our results identified *NOX4*-dependent ROS as a major contributor to promote DF-induced *HIF-1α* stabilization. The data are consistent with a previous study showing that oscillatory flow increases *NOX4* expression and ROS production in cultured bovine aortic endothelial cells (*Hwang et al., 2003*). In agreement with these in vitro investigations, we detected increased *NOX4* expression and ROS generation, accompanied by elevated *HIF-1α*, in vivo in endothelia isolated from the atherosusceptible swine aortic arch. ROS are known to stabilize *HIF-1α* even in normoxic conditions (*Bonello et al., 2007*). *HIF-1α* stabilization has also been shown to occur in stretch-based assays by stretch-induced inhibition of succinate dehydrogenase, and is thought to mediate epithelial protection during ventilator-induced lung injury (*Eckle et al., 2013*, *2014*). While it is possible that accumulation of mitochondrial TCA substrates contributes to *HIF-1α* stabilization under DF, flow by itself is unlikely to cause mechanical stretch in either the in vitro assays or in the aorta in vivo, as UF is more likely than DF to cause stretch (UF has 10-fold higher shear stress compared to DF). *HIF-1α* mRNA is also thought to increase after prolonged stretch in rat skeletal muscle capillary beds (*Milkiewicz et al., 2007*; *Milkiewicz and Haas, 2005*) and the inferior vena cava (*Lim et al., 2011*). However, we found no transcriptional differences between UF and DF (*Figure 2C*).

Another major finding of the present study is the demonstration that DF-induced endothelial metabolic reprogramming contributes to increased inflammatory gene expression. DF-induced expression of pro-inflammatory genes was markedly attenuated with the knockdown of *SLC2A1*, *PDK1*, or *HIF-1α* (loss-of-function), and increased by *HIF-1α* overexpression (gain-of-function). Corroborating our in vitro results, we detected increased *HIF-1α*, *SLC2A1*, and *PDK1* expression in vivo in swine aortic arch ECs. These results are in agreement with the reported increase in the expression of *HIF-1α* in carotid artery plaques (*Vink et al., 2007*; *Sluimer et al., 2008*). Aortic arch ECs have exposed to DF in vivo also been shown to exhibit *NF-κB* activation (*Hajra et al., 2000*; *Fang et al., 2010*). Although the underlying molecular mechanisms remain to be elucidated, one possibility is that increased glycolysis promotes endothelial inflammation via lactate-induced activation of *NF-κB* signaling in a manner described in immune cells such as macrophages (*Végran et al., 2011*; *Covarrubias et al., 2015*). However, it is important to point out that we cannot rule out the possibility that *HIF-1α* promotes vascular inflammation by additional pathways that are independent of metabolic changes. For example, our RNA-seq analyses demonstrated that inhibition of *HIF-1α* in DF-treated ECs rescued the expression of anti-inflammatory molecules *KLF2*, *KLF4*, and *PPAP2B* (a coronary arterial disease candidate gene identified by genome-wide association studies)

(*SenBanerjee et al., 2004*; *Wu et al., 2015*). While we did not study the role of *KLF2*, Doddaballa-pur et al. showed that *KLF2* is required for UF-induced reduction in the expression of glycolytic genes in human umbilical vein endothelial cells (*Doddaballapur et al., 2015*). Importantly, Inflamma-tory stimuli such as *TNFα* can by itself reduce the UF-responsiveness of *KLF2* (*Huang et al., 2017*); it remains to be seen if metabolic changes might also play a role under such stimuli. Thus, differential flow waveforms may amplify EC metabolic phenotypes (and subsequent pro- or anti-inflammatory states) by providing transcriptional mechanisms to increase glycolysis and reduce respiration under DF and to reduce glycolysis and increase respiration under UF.

Lesion-targeted therapies may fill the current treatment gap related to systemic risk factor man-agement (*Kuo et al., 2014*). Our in vitro and in vivo results suggest spatial inhibition of glycolytic metabolism and *HIF-1α* signaling may be a suitable approach for future arterial wall-based therapies that target disturbed flow-activated endothelium associated with focal atherosclerosis. Consistent with the results demonstrating *HIF-1α* inhibition reduces the disturbed flow-induced inflammation, endothelium-deletion of *HIF-1α* was recently shown to significantly ameliorate atherosclerotic bur-den in $apoe^{-/-}$ mice (*Akhtar et al., 2015*).

In summary, while studies suggest inhibition of glycolysis as a potential therapy to reduce angio-genesis (*Schoors et al., 2014*), we propose that there may be a role in promoting the athero-protec-tive flow phenotype in order to reverse EC activation such as increased vascular permeability and inflammation. Further investigations in metabolic effects of athero-protective flow might provide useful insight into the pathways that may be targeted to promote mitochondrial health and reduce athero-susceptibility. Here, we described that metabolic plasticity of endothelium is dynamically and reversibly regulated by hemodynamics suggesting a critical role for mechanical forces in actively mediating metabolic changes that drive endothelial functions in health and disease.

## Materials and methods

### Primary culture of human aortic endothelial cells

Human aortic endothelial cells (HAECs) were purchased from Lonza (Allendale, NJ) (CC-2535, lot number 0000305906 or 0000297640 or 0000336393). The primary cells were isolated from donated human tissue (male, age 27, female, 54, and male, age 22, respectively). The cells were tested for mycoplasma. The cells were verified to be endothelial cells through cell morphology (alignment with flow) and presence of CD31. There was no alpha actin expression. For experiments, cells were grown in EGM-2 supplemented with SingleQuots from Lonza (CC-3156 and CC-4176) and Antibiotic-Anti-mycotic from Gibco (Grand Island, NY) (15240062). For flow experiments, cells were plated in 6-well plates at $4 \times 10^5$ cells/well, and after 24 hr, dextran (Sigma-Aldrich, St. Louis, MO, 31392) was added to media to final concentration of 4%. All primary cultures were used from passage 6 to 10.

### Cell culture of THP-1 cells

THP-1 (human monocyte) cells were a gift from the Sperling lab at the University of Chicago. Cells were grown in RPMI media (ThermoFisher, Waltham, MA) and 10% FBS (Biowest USA, Riverside, MO).

### Application of flow

A flow device consisting of a computerized stepper motor UMD-17 (Arcus Technology, Livermore, CA) and a 1° tapered stainless steel cone was used to generate the physiologically-relevant shear stress pattern (*Wu et al., 2015*). The flow device was placed in a 37°C incubator with 5% CO2. HAECs at 100% confluence, maintained in EGM2- medium containing 4% dextran in 6-well plates, were subjected to unidirectional steady flow (UF) or disturbed flow (DF) for 24–72 hr before cells being harvested (*Dai et al., 2004*). Static cells used the same dextran-containing media above and did not utilize the flow devices.

### Cell lysis and western blotting

Cells were lysed with buffer containing 8M deionized urea, 1% SDS, 10% glycerol, 60 mM Tris-HCl, 5% betamercaptoethanol (all chemicals from Sigma-Aldrich). The lysates are passed through an insu-lin syringe three times and resolved on 4–12% Bis-Tris gels (Invitrogen) and transferred to a 0.2 $\mu$m

PVDF membrane before blocking in either 5% non-fat milk in Tris-buffered saline and 0.1% Tween-20 (TBST) or 5% bovine serum albumin in TBST. Blots were then incubated in primary antibodies as described below.

## Quantitative PCR

RNA as isolated form cells using Trizol and the Zymo Direct-zol RNA MiniPrep kit and reverse transcribed using High-Capacity cDNA Reverse Transcription Kit (ThermoFisher). Quantitative mRNA expression was determined by real-time RT-PCR using SYBR Green MasterMix (Roche, Indianapolis, IN). The following primer sequences were used (IDT, Coralville, IA):

SLC2A1- 5'-GAACTCTTCAGCCAGGGTCC-3', 5'-ACCACACAGTTGCTCCACAT-3'
HK2- 5'-GCTTGGAGCCACCACTCACCC-3', 5'-AGCCAGGAACTCTCCGTGTTCTGT-3'
IL8, 5'-TGTGCCTTGGTTTCTCCTTT-3' 5'-GCTTCCACATGTCCTCACAA-3'
CCL2, 5'-AGCAGCAAGTGTCCCAAAGA-3' 5'-TTGGGTTTGCTTGTCCAGGT-3'
VCAM1, 5'-GATACAACCGTCTTGGTCAG-3' 5'-TAATTCCTTCACATAAATAAACCC-3'
ND1 5'-AGCCCTACTCCACTCAAGCA-3' 5'-GCTGCGAACAGAGTGGTGAT-3'
ND5 5'-GCCCTACTCCACTCAAGCAC-3' 5'-TGAAGAAGGCGTGGGTACAG-3'
HIF-1$\alpha$ 5'-GGCGCGAACGACAAGAAAAA-3' 5'-GTGGCAACTGATGAGCAAGC-3'
PDK1 5'- CCAGGACAGACAATACAAGTGGT-3' 5' GAATCGGGGGATAAACGCCT-3'
EPAS1 5'-GCGCACCTCGGACCTTCA-3' 5'- TCTCCGAGCTACTCCTTTTCTTC-3'
EGLN3 5'-CACAGCGAGGGAATGAACCT-3' 5'-TCCTGCTGTTAAGGCTTCCG-3'
VEGFA 5'-CTCTACCTCCACCATGCCAA-3' 5'-GCATGGTGATGTTGGACTCC-3'
KLF2 5'-GAACCCATCCTGCCGTCCTT-3' 5'-CACGCTGTTGAGGTCGTCG-3'
KLF4 5'-ATCTCAAGGCACACCTGCG-3' 5'- CCTGGTCAGTTCATCTGAGCG-3'
GAPDH 5'- TGCACCACCAACTGCTTAGC-3' 5'- GGCATGGACTGTGGTCATGAG-3'
ACTB 5'- TCCCTGGAGAAGAGCTACGA-3' 5'- AGGAAGGAAGGCTGGAAGAG-3'
UBB 5'- ATTTAGGGGCGGTTGGCTTT-3' 5'- TGCATTTTGACCTGTTAGCGG-3'

## Glycolysis stress test, mitochondrial stress test, and mitochondrial substrate utilization

Glycolytic rates were measured using the XF$^e$24 Extracellular Flux Analyzer (Seahorse Bioscience, North Billerica, MA). After siRNA, chemical inhibitors, or flow experiments, HAECs were collected and seeded at a density of $4 \times 10^4$/well on Seahorse plates and allowed to adhere for 4 hr in a standard incubator. Cells were next equilibrated with XF Base media (Seahorse) at 37°C for one hour in an incubator lacking $CO_2$. Glycolysis stress test was performed by sequential treatments with glucose (10 mM), oligomycin (1.0 μM) and 2-DG (100 mM)). Mitochondrial stress test was performed by sequential treatments with oligomycin (1.0 μM), carbonyl cyanide-4-(trifluoromethoxy)phenylhydrazone (FCCP) (1.0 μM), and rotenone/antimycin A (1.0 μM each). For determination of oxidative phosphorylation substrate utilization, cells were equilibrated in XF Base media supplemented with glucose (5 mM) and glutamine (2 mM). For glucose utilization determination, UK5099 (2 μM) and then BPTES (3 μM)/Etoxomir (4 μM) (4 M) or BPTES (3 μM)/Etoxomir (4 μM) and then UK5099 (2 μM) were added sequentially. Dichloroacetate (DCA) at 4 mM was maintained in the media at all times for DCA experiment. All chemicals form Sigma-Aldrich.

## siRNA knockdowns

HAECs ($4 \times 10^5$ cells) were transfected with an siRNA (50 nmol) using RNAiMAX (Life Technologies, Carlsbad, CA) as described by the manufacturer. Media was exchanged the following day for flow media. siRNA products are obtained from Qiagen (Venlo, The Netherlands) and the siRNAs used are as follows:

Non-targeting siRNA sense: 5'-UUCUCCGAACGUGUCACGUdTdT 3' 1027310
Non-targeting siRNA antisense: 5'-ACGUGACACGUUCGGAGAAdTdT 3' 1027310
HIF-1$\alpha$ siRNA sense: 5'-GAAGAACUAUGAACAUAAATT-3' SI02664053
HIF-1$\alpha$ siRNA antisense: 5'-UUUAUGUUCAUAGUUCUUCCT-3' SI02664053
NOX4 siRNA sense: 5'-CCAGGAGAUUGUUGGAUAATT-3' SI02642500
NOX4 siRNA antisense: 5'-UUAUCCAACAAUCUCCUGGTT-3' SI02642500
PDK1 siRNA sense: 5'- CGAACUAGAACUUGAAGAATT-3' SI00605752

*PDK1* siRNA antisense: 5'-UUCUUCAAGUUCUAGUUCGGG-3' SI00605752
*SLC2A1* siRNA sense: 5'-CCACGAGCAUCUUCGAGAATT-3' SI03089401
*SLC2A1* siRNA antisense: 5'-UUCUCGAAGAUGCUCGUGGAG-3' SI03089401

## HIF-1α overexpression via transfection with in vitro transcripts

In vitro transcription for wild type (T7-*HIF-1α*) or mutated *HIF-1α* (T7-*mHIF-1α*) mRNA transcripts were performed using mMESSAGE mMACHINE T7 Ultra Kit (Life Technologies) following manual instructions. HAECs were transfect with 40 ng of T7-*KLF2* or T7-ctrl with Lipofectamine Messenger-MAX (Life Technologies) for 6 hr according to manufacturer's instructions.

## Generation of triple mutant *mHIF-1α*

Human *HIF-1α* (*hHIF-1α*) expression plasmid was purchased from GeneCopoeia (Rockville, MD). Mutagenesis of P402, P564 and N803 in *hHIF-1α* open reading frame was performed using Quik-Change Multi Site-Directed Mutagenesis Kit (Agilent (Newport, DE)) according to the manufacturer's protocol. Mutations were confirmed by DNA sequencing. Primer sequences for mutagenesis:

*hHIF-1α*_P402A_F: 5'-TAACTTTGCTGGCCGCTGCCGCTGGAGACAC-3'
*hHIF-1α*_P564G_F:    5'-GATTTAGACTTGGAGATGTTAGCTGGATATATCCCAATGGATGATGAC TTC-3'
*hHIF-1α*_N803A_F: 5'-CTGACCAGTTATGATTGTGAAGTTGCTGCTCCTATACAAGGCAG-3'

## HIF-1α overexpression via transfection with in vitro transcripts

Wild type *HIF-1α* and *mHIF-1α* plasmids were used to generate the DNA templates for in vitro transcripts with the primers listed below. In vitro transcription for wild type (T7-*HIF*) or *mHIF-1α* (T7-*mHIF-1α*) mRNA transcripts was performed using mMESSAGE mMACHINE T7 Ultra Kit (Life Technologies) following manual instructions. HAECs were transfected with T7-*HIF-1α* or T7-*mHIF-1α* mRNA transcripts using Lipofectamine MessengerMAX (Life Technologies) according to manufacturer's instructions.

hT7_F: 5'-CCA CTG CTT ACT GGC TTA TCG-3'
*hHIF-1α* _stop: 5'-TCA GTT AAC TTG ATC CAA AGC TCT G-3'

## CellRox and MitoTracker assay

A 6-well plate had its bottom sawed off and hand-filed down. A #1 microscope coverslip (Ted-Pella, Redding, CA) sufficient to cover the 35 mm$^2$ area was adhered with aquarium seal and allowed to dry over-night. The plates were re-sterilized with ethanol and UV for 48 hr prior to use. 1 hr before cell plating, 0.1% gelatin in PBS was pipetted onto the cover glass and incubated at 37°C. HAECs were then plated onto the coverglass overnight before flow experiments. After flow experiments, the media was changed to EGM-2 with either 5 mM CellRox Orange (ThermoFisher) or 500 $\mu$M Mito-Tracker Deep Red (ThermoFisher) and placed in the incubator for 30 min. The media was changed again for EGM-2 and the cells imaged immediately. A standard mercury lamp and TRITC (CellRox) or Cy5 (MitoTracker) filter set was used on an inverted microscope (Zeiss Axio 200).

## 2-NBDG uptake assay

HAECs were plated at $4 \times 10^5$ cells/well overnight before either UF or DF flow for 48 hr. Immediately after flow, cells were washed with warm PBS three times before incubating with Krebs-Ringer-Phosphate-HEPES (KPRH) buffer (20 mM HEPES, 5 mM $KH_2PO_4$, 1 mM $MgSO_4$, 1 mM $CaCl_2$, 136 mM NaCl, 4.7 mM KCl, pH 7.4) with 2% BSA for 1 hr. Cells were again washed with PBS three times and incubated in 1 mM 2-NBDG and PBS for 30 min. The cells were again washed with warm PBS three times and KPRH/BSA was added to the cells. The cells were immediately imaged on a Nikon Ti-Eclipse (Tokyo, Japan) microscope at 20x using a FITC filter set. 20 random locations were imaged in both epi-fluorescence and brightfield. Fluorescence brightness was quantified using a custom MATLAB script, deposited at the journal website. Reagents for the assay are from Abcam #136955 (Cambridge, MA).

## ATP assay ((Abcam, ab113849) luminescence ATP detection assay kit)

HAECs were plated at $4 \times 10^5$ cells/well overnight before either UF or DF flow for 48 hr. Cells were washed with warm PBS three times before trypsinized and re-plated at 20,000 cells/well (four replicates per condition) in 90 µL of EGM2 media in a flat white-bottom plate and allowed to settle for 4 hr at 37 C. Next, either 10 µL PBS, 10 µL of rotenone/antimycin (final concentration 1 µM each) or 10 µL 2-deoxyglucose (final concentration 50 mM) were added to the cells, and incubated at 37 C for 30 min. The cells were then assayed according to the manufacturer's protocol.

## Porcine aorta samples

Porcine aortas were obtained from a local slaughterhouse (Ruwaldt Packing Co. (Hobart, IN)). Three-year old male pigs destined to enter the food supply were sacrificed and submerged in boiled in water for 10 s per industry protocol. They were rapidly dissected and the aortas removed. The aortas were cut open and washed quickly with cold PBS before gently passing a #10 scalpel across the lumen. 100 µL of lysis buffer was quickly added to the cell scraping and the samples frozen on dry ice. The entire process took less than 10 min. Swine protein lysates were immunoblotted for smooth muscle cell marker SM22-alpha to verify the endothelial purity. Endothelial purity was detected at greater than 95%.

## Mouse aorta imaging

All animal care and treatment procedures were approved by the University of Chicago Institutional Animal Care and Use Committee. C57BL/6J mice (Jackson Laboratory, Bar Harbor, ME) were anesthetized with trimobromoethanol (2.5%, 17.5 µL/kg, intraperitoneal injection) (Sigma). As soon as their reflexes were blunted, as assessed by hindleg pinching, the mice were supinated and their limbs taped to a dissection board. Mice were sprayed with 70% ethanol prior to dissection: the thoracic cage was opened and the heart visualized. 5 mL of warm PBS was flushed through the left ventricle with a 25 gauge butterfly needle with the IVC open. Next, 4% paraformaldehyde (PFA) (Sigma) in PBS was perfused through the left ventricle at a rate of 2 mL/min for 15 min. At the end of perfusion, 5 mL of PBS was again used to wash out any remaining PFA. The aorta was then removed from the thorax and cleaned. The heart and aorta were removed *en bloc* and the descending thoracic aorta separated from the arch, below the left subclavian artery.

For immunofluorescence, the aortas were opened along the dorsal aspect before pinning to a rubber board. The aortas were permeablized with 0.1% Triton-X 100/PBS for 10 min and washed with 5% BSA and tris-buffered saline with TBST three times, prior to incubation with 1:100 *HIF-1α* antibody (rabbit Ig) or 1:50 NOX4 antibody (rabbit Ig) in 5% BSA and TBST overnight at 4C, together with 1:100 CD31 antibody (rat Ig). After primary antibody incubation, the samples were again washed with 5% BSA and TBST three times and incubated with 1:1000 goat anti-rabbit IgG Alexa 488 (ThermoFisher, A11034), 1:1000 chicken anti-rat IgG Alexa 594 (ThermoFisher, A21471) and 1:2000 Hoechst (ThermoFisher, 62249) for 1 hr at room temperature. The samples were again washed with 5% BSA and TBST three times prior to mounting on a coverslip for imaging. Imaging was performed on a spinning disk confocal microscope.

## Other reagents and antibodies

Dichloroacetate was purchased from Sigma-Aldrich (347795). Used at 4 mM final concentration. EUK134 was purchased from Sigma-Aldrich (SML0743) and used at 1 mM final concentration. NEMO-binding domain peptide (480025) from Millipore. Rotenone and antimycin were purchased from Sigma-Aldrich. Primary antibodies used were *HIF-1α* (1:500 in 5% non-fat milk and TBST for Western, 1:100 in 5% BSA and TBST for immunofluorescence, Cayman, Ann Arbor, MI, 10006421, RRID:AB_10099184), *SLC2A1* (1:1000 in 5% BSA and TBST, ProteinTech, Rosemont, IL, 21829–1-AP, RRID:AB_10837075), *HK2* (1:1000 in 5% BSA and TBST, Cell Signaling, Danvers, MA, C64G5, RRID: AB_2232946), *β*-actin (1:5000 in 5% BSA and TBST, Abcam, ab6276, RRID:AB_2223210), *PDK1* (1:1000 in 5% non-fat milk and TBST, Cell Signaling, C47H1, RRID:AB_1904078), *NOX4* (1:500 in 5% milk and TBST for Western, 1:50 in 5% BSA and TBST for immunofluorescence, Santa Cruz, Dallas, TX, sc-30141, RRID:AB_2151703), *CD31* (BD, Franklin Lakes, NJ 550274, RRID:AB_393571), *EPAS1* (1:1000 in 5% BSA and TBST, Novus Biologicals, Littleton, CO, NB100-122, RRID:AB_10002593). Secondary antibodies for Western blotting: Goat anti-mouse IgG, HRP conjugate (1:5000 in 5% non-

fat milk and TBST, Calbiochem, 401253, RRID:AB_437779). Anti-rabbit IgG, HRP conjugate (1:3000 in 5% non-fat milk and TBST, Cell Signaling, 7074S, RRID:AB_2099233).

## Pyruvate dehydrogenase (*PDH*) activity (Abcam, ab109882)

HAECs were plated at $4 \times 10^5$ cells/well in a 6-well plate overnight before either UF or DF flow for 48 hr before proceeding with the protocol according to the manufacturer. Briefly, the cells were trypsinized and resuspended in sample buffer. 1/10 vol of detergent was added and the sample mixed and stored on ice for 10 min. The sample was then centrifuged and the supernatant collected, and the protein concentration measured with a detergent-compatible BCA protein assay (Pierce), according to standard protocol. Sample volumes were adjusted so that the total amount of protein loaded onto dipsticks was the same. The *PDH* dipsticks were first blocked and before wicking sample. The dipsticks were then washed with sample buffer before the addition of activity buffer. The reaction was allowed to proceed for 1 hr before washing with deionized water and drying. The dipsticks were the imaged by a flatbead scanner and quantified with ImageJ. The final intensity was normalized to total protein.

## Glucose uptake assay (Abcam, ab136955)

HAECs were plated at $4 \times 10^5$ cells/well in a 6-well plate overnight before either UF or DF flow for 48 hr. The cells were removed from the flow devices and washed with PBS three times. The cells were then starved in KPRH/BSA buffer for 40 min before washing again in PBS three times. Samples were then incubated in either 10 mM 2-deoxyglucose (2-DG) or PBS (control) for 20 min. The HAECs were then washed with PBS three times before trypsinization and cell counting. The final cell number was normalized across all experiment and control samples. The samples were pelleted and resuspended in PBS with addition of 80 $\mu$L Extraction Buffer and vigorously mixed. After on freeze/thaw cycle on dry ice, the the samples were incubated at 85$^\circ$C for 40 min. The lysates were cooled on ice for 5 min and then 10 $\mu$L of Neutralization Buffer was added. The samples were mixed and centrifuged. The supernatants were then assayed according to the manufacturer protocol, and final 412 nm absorbance reading normalized to non-2DG treated controls. The 412 nm absorbance was compared against a standard curve according to manufacturer's protocol.

## Leukocyte adhesion assay

HAECs were subjected to 50 nM siRNA targeted against *HIF-1$\alpha$* or control for 24 hr, prior to DF for 48 hr. 1 hr prior to removal from flow, THP-1 cells at 5-fold number of HAECs were pelleted and resuspended in 600 $\mu$L serum-free RPMI (900 rpm, 5 min, room temperature). The THP-1 cells were then incubated with 5 $\mu$M Calcein AM dye (ThermoFisher) for 30 min at 37 C. The cells were then pelleted and resuspended in 600 $\mu$L (900 rpm, 5 min, room temperature) serum free RPMI. After removal of flow devices, the HAEC flow medium was replaced with regular EGM2 (1 mL). 100 $\mu$L of labeled THP-1 cells was then added to the HAEC cells and let incubate at 37 C for 1 hr, with gentle rocking every 30 min. The HAEC+THP-1 cells were then washed with warm PBS five times, 2 mL per well. Fluorescence of the Calcien AM dye was then measured on a Cytation 3 (Biotek, Winooski, VT) device in area scanning mode, with gain of 80, and excitation 492 nm, and emission 550 nm.

## RNA sequencing

Briefly, quality of reads was assessed using fastQC. Reads were aligned to GENCODE hg38.p2 reference genome using Tophat2 version 2.1.1. Transcripts were assembled using the bam files from Tophat2 using Cufflinks version 2.1.1. The transcript files from cufflinks were merged using cuffmerge. Cuffquant was used to estimate abundances, prior to analysis by cuffdiff to estimate differential gene expression. The cut off for differentially expressed genes was an FDR of 0.05 (*Trapnell et al., 2012*). The complete unedited RNA-seq datasets are available at doi: https://doi.org/10.5281/zenodo.260122 (*Wu et al., 2017b*) and doi: https://doi.org/10.5281/zenodo.260120 (*Wu et al., 2017a*). The list of differentially expressed genes after processing has been uploaded to the publisher's website.

## Gene set enrichment analysis (GSEA)

Genes used for GSEA (*Subramanian et al., 2005*; *Mootha et al., 2003*) were all genes that were tested by cuffdiff, regardless of whether or not they were differentially expressed in either condition. All other genes that were not tested were deemed too lowly expressed or detected to be relevant. The gene set collection used was the h.all.v5.1.symbols.gmt [Hallmarks] gene sets database. GSEA performed 1000 permutations. Phenotype labels corresponded to the conditions of the experiment. Did not collapse dataset to gene symbols. Permutation type was gene_set. Chip platform was ftp://gseaftp.broadinstitute.org/pub/gsea/annotations/GENE_SYMBOL.chip. Enrichment statistic was weighted. Metric for ranking genes was Signal2Noise. The gene list sorting mode was real and done in descending ordering mode. The max size was 500 and the minimum was 15 for gene sets. Cutoff for significant gene sets was an FWER less than 1.0. In this study, we selected to report the ten most significant gene sets.

## Ingenuity pathway analysis

Ingenuity Pathway Analysis (IPA, Redwood City, CA), www.ingenuity.com) is a system that transforms a list of genes into a set of relevant networks based on extensive records maintained in the Ingenuity Pathways Knowledge Base (*Calvano et al., 2005*). Highly-interconnected networks are predicted to represent significant biological function (*Ravasz et al., 2002*).

IPA Upstream Regulator Analytics was used to computationally predict the putative upstream transcriptional regulators that contribute to the gene expression changes in HAECs under athero-relevant hemodynamics. 3757 differentially expressed genes (DEGs) identified by the RNA-seq at a false discovery rate cut off value of q < 0.05 were uploaded to the IPA software. The analysis was conducted based on prior knowledge of expected effects between transcriptional regulators and their target genes stored in the Ingenuity Knowledge Base. The putative upstream regulators were ranked by an overlap p-value which calls likely upstream regulators based on significant overlap between dataset genes and known targets regulated by a transcriptional regulator. The activation z-score statistic is a weighted sum of activating and inhibiting interactions on a given gene set (*Krämer et al., 2014*).

## DAVID

We used DAVID (*Huang et al., 2009a*, *Huang et al., 2009b*) version 6.7. Only differentially expressed genes were included and were separated if they had a positive or negative fold change from the RNA sequencing analysis. Summary results used were from GOTERM_BP_FAT. The top ten most significant gene ontology (GO) terms were used and the p-values were reported.

## Metascape

Gene list of differentially expressed genes was used at www.metascape.org. Express analysis was used.

## Statistics

The data were analyzed in Prism 7 (GraphPad Software Inc, La Jolla, CA) or Excel. All data are shown as mean ± standard error of the mean (SEM). All data are in two groups. We explored individual differences with two-tailed Student's t test. Statistical significance was defined as p<0.05. A *P* value of less than 0.05 was considered significant. Due to experimental constraints, only eight biological replicates (four control, four experiment) are able to be run simultaneously with flow-devices. All plotted experiments are biological replicates except for Seahorse metabolic experiments, which are technical replicates. Seahorse experiments were run at least twice. All p-values are available at the publisher's website.

## Study approval

All procedures were in accordance with National Institutes of Health guidelines, and the use of vertebral animal related tissues obtained from outside the University of Chicago was approved by the Institutional Animal Care and Use Committee.

## Other source data

RNA-seq differentially expressed genes for either UF or DF, and for DF with siRNA directed towards *HIF-1α* can be found online at the journal website. The complete unedited RNA-seq datasets are available at Flow transcriptome of human aortic endothelial cells, doi: https://doi.org/10.5281/zenodo.260122 (*Wu et al., 2017b*) and *HIF-1α* knockdown under disturbed flow in human aortic endothelial cells, doi: https://doi.org/10.5281/zenodo.260120 (*Wu et al., 2017a*).

Source code for image analysis is available as *Source code 1*, written in MATLAB code.

## Acknowledgements

Information about direct funding can be found at the journal website. No funding sources were involved in the study design, data collection, interpretation, or the decision to submit the work for publication. We also acknowledge the support of the University of Chicago Microscopy Core and Genomics Facility and thank Dr. Navdeep S Chandel (Northwestern University) and Dr. Godfrey Getz (University of Chicago) for careful review of our manuscript.

## Additional information

### Funding

| Funder | Grant reference number | Author |
| --- | --- | --- |
| National Institutes of Health | T32HL007605 | David Wu<br>Myung-Jin Oh |
| American Heart Association | 15POST255900003 | Recep Nigdelioglu |
| National Institutes of Health | F32HL134288 | David Wu |
| National Institutes of Health | R21ES025644 | Gökhan M Mutlu |
| National Institutes of Health | K01AR066579 | Robert B Hamanaka |
| National Institutes of Health | R01ES015024 | Gökhan M Mutlu |
| National Institutes of Health | P01HL090554 | Nanduri R Prabhakar |
| National Institutes of Health | R00HL103789 | Yun Fang |
| American Heart Association | BGIA7080012 | Yun Fang |

The funders had no role in study design, data collection and interpretation, or the decision to submit the work for publication.

### Author contributions

DW, Conceptualization, Data curation, Formal analysis, Investigation, Methodology, Writing—original draft, Writing—review and editing; R-TH, RN, AYM, Investigation; RBH, Conceptualization, Methodology, Writing—review and editing; MK, LW, GD, Investigation, Methodology; M-JO, C-HK, Data curation, Investigation; MC, Conceptualization, Investigation, Writing—review and editing; NRP, YF, Conceptualization, Resources, Supervision, Funding acquisition, Methodology, Writing—original draft, Writing—review and editing; GMM, Conceptualization, Resources, Supervision, Funding acquisition, Investigation, Writing—original draft, Writing—review and editing

### Author ORCIDs

David Wu, http://orcid.org/0000-0003-3162-3238
Robert B Hamanaka, http://orcid.org/0000-0002-8909-356X
Cheng-Hsiang Kuo, http://orcid.org/0000-0002-4885-9020
Yun Fang, 0000-0003-4597-3095
Gökhan M Mutlu, http://orcid.org/0000-0002-2056-612X

### Ethics

Animal experimentation: All procedures were in strict accordance with the recommendations in the Guide for the Care and Use of Laboratory Animals of the National Institutes of Health. The use of

vertebrate animal tissues was approved by the Animal Care and Use Committee of the University of Chicago (Permit # 72281). The use of vertebrate animal tissues obtained from outside the University of Chicago was approved by the Animal Care and Use Committee of the University of Chicago (Permit # 72500).

## Additional files

### Supplementary files
• Source code 1. Image analysis algorithm.

### Major datasets
The following datasets were generated:

| Author(s) | Year | Dataset title | Dataset URL | Database, license, and accessibility information |
|---|---|---|---|---|
| Wu D, Huang R-T, Hamanaka RB, Krause MD, Kuo C-H, Nigdelioglu R, Meliton AY, Witt L, Dai G, Civelek M, Prabhakar NR, Fang Y, Mutlu GM | 2017 | Flow transcriptome of human aortic endothelial cells | https://doi.org/10.5281/zenodo.260122 | Publicly available at Zenodo.org under a Creative Commons Attribution 4.0 License |
| Wu D, Huang R-T, Hamanaka RB, Krause MD, Kuo C-H, Nigdelioglu R, Meliton AY, Witt L, Dai G, Civelek M, Prabhakar NR, Fang Y, Mutlu GM | 2017 | HIF-1$\alpha$ knockdown under disturbed flow in human aortic endothelial cells | https://doi.org/10.5281/zenodo.260120 | Publicly available at Zenodo.org under a Creative Commons Attribution 4.0 License |

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
