## [Decision Letter]

Thank you for submitting your article "HIF-1α is required for disturbed flow-induced metabolic reprogramming in vascular endothelium" for consideration by *eLife*. Your article has been favorably evaluated by Philip Cole (Senior Editor) and three reviewers, one of whom is a member of our Board of Reviewing Editors. The following individual involved in review of your submission has agreed to reveal his identity: Mukesh Jain (Reviewer #3).

The reviewers have discussed the reviews with one another and the Reviewing Editor has drafted this decision to help you prepare a revised submission.

Summary:

Wu et al. show that endothelial metabolic changes in response to disturbed flow (DF), typically seen in athero-susceptible areas, induce glycolysis and reduce mitochondrial respiratory capacity in a HIF-1 dependent manner. Furthermore, they show that HIF levels are increased in athero-susceptible regions of the aorta. Blockade of HIF or key metabolic targets abrogates this effect and reduces endothelial inflammation. As an underlying mechanism, the authors propose an NOX4-dependent increase in reactive oxygen species (ROS), which block HIF-1α degradation and thereby increase HIF-1α signaling. Blockade of HIF-1α-driven glycolytic genes is further demonstrated to reduce endothelial inflammatory gene expression, which leads the authors to the conclusion that metabolic reprogramming plays an important role in vascular inflammation.

The information here adds important insights to the growing appreciation regarding the importance of EC metabolism in vascular biology. It focuses on an interesting and timely topic – the role of endothelial metabolic adaptation in vascular responses to different types of flow.

Essential revisions:

The mechanism by which altered EC metabolism affects the inflammatory gene expression remains a black box with few hints how the former leads to the latter. Moreover, one would like to know which aspect of EC metabolism drives inflammation; increased glycolysis, decreased OXPHOS or both? The authors could at least try to assess the functional consequences of altered metabolism in vitro, e.g. by analyzing leukocyte adhesion or other surrogate parameters of endothelial inflammation.

Many of the effects observed for HIF have, based on studies in EC or other cell types, also been attributed to NFkB. Further, there is a long-standing appreciation that NFkB activity is increased in DF as well as athero-susceptible regions. Is NFkB activity increased under DF? Is this activity important for HIF-1 induction (as has been suggested in other cell types) or are these independent pathways?

The manuscript would also profit from additional evidence indicating that the NOX4-ROS axis leads to significant HIF1 stabilization and activation of the HIF-1 pathway in vivo. The authors could consider examining Hif-1 staining, Hif-1 reporter mice, which are available (http://www.pnas.org/content/103/1/105.full), or mice with endothelial deletion of Hif-1 (Covarrubias, Aksoylar and Horng, 2015).

---

## [Author Response]

*Essential revisions:*

*The mechanism by which altered EC metabolism affects the inflammatory gene expression remains a black box with few hints how the former leads to the latter. Moreover, one would like to know which aspect of EC metabolism drives inflammation; increased glycolysis, decreased OXPHOS or both? The authors could at least try to assess the functional consequences of altered metabolism* in vitro*, e.g. by analyzing leukocyte adhesion or other surrogate parameters of endothelial inflammation.*

We concur with the editors/reviews that it is important to determine the functional consequences of altered EC metabolism. We assessed whether manipulating metabolism affects leukocyte adhesion by subjecting HAECs to siRNA targeted against the HIF-1α and subjecting them to disturbed flow for 48 hours, then performing a leukocyte adhesion assay. Consistent with previously studies, (1, 2), disturbed flow causes increased leukocyte adhesion relative to unidirectional flow. Moreover, we found that knocking down HIF-1α significantly reduced THP-1 adhesion compared to scrambled control (new Figure 6—figure supplement 1).

As to whether increased glycolysis or reduced OXPHOS drives inflammation, the analysis remains challenging. A Crabtree effect exists in endothelial cells, wherein changes in glycolysis (or surrogate extracellular acidification rate (ECAR)) are automatically reflected as an opposing change in mitochondria oxidation (reflected in the oxygen consumption rate (OCR)), and vice-versa (3). Thus, it is difficult to “clamp” either glycolysis or oxidative phosphorylation and let the other vary independently. We have tested this in our HAEC cells and have found it to be the case. While we do not include this data in the manuscript, we provide it as Figure 8. As you can see, when glucose is added to HAECs, the ECAR jumps as expected (black line, arrow A, left panel). Simultaneously, the OCR decreases (black line, arrow A, right panel). Similarly, adding glutamine, which enters the TCA cycle via conversion to glutamate and subsequent conversion to α-ketoglutarate, reduces ECAR (red line, arrow A, left panel). There is minimal change in OCR (red line, arrow A, right panel). After initial addition of glucose, subsequent addition of glutamine reduces ECAR (black line, arrow B, left panel) and modestly increases OCR (black line, arrow B, right panel). After initial addition of glutamine, subsequent addition of glucose increases ECAR (red line, arrow B, left panel), and reduces OCR (red line, arrow B, right panel).

Author response image 1.**DOI:**
http://dx.doi.org/10.7554/eLife.25217.039

In summary, our results demonstrate a Crabtree effect in HAECs and the control of either ECAR or OCR independently in these cells is difficult.

With these caveats and limitations above, we attempt to answer the question whether reducing one or the other of glycolysis or oxphos reduces inflammation. We previously demonstrated that siRNA targeted knockdown of SLC2A1 (GLUT1, the major glucose transporter upregulated in disturbed flow) reduced markers of inflammation (Figure 6). We now also provide extracellular flux analysis confirming that siRNA against SLC2A1 reduces glycolysis (new Figure 6—figure supplement 2), and evidence that knocking down SLC2A1 with siRNA reduces the SLC2A1 mRNA and protein content of cell lysates, as shown in new Figure 6—figure supplement 2.

To extend the metabolic analysis, we treated HAECs with rotenone and antimycin, which together block complex I and III; no electrons are donated to the electron transport chain ETC and OCR is reduced (Figure 1). We show that inhibiting the ETC increases inflammation under UF (Figure 6—figure supplement 3). These results suggest that reducing electron transport increases endothelial inflammation.

Changes were made in the section “Disturbed flow-induced metabolic reprogramming is required for endothelial activation”.

*Many of the effects observed for HIF have, based on studies in EC or other cell types, also been attributed to NFkB. Further, there is a long-standing appreciation that NFkB activity is increased in DF as well as athero-susceptible regions. Is NFkB activity increased under DF? Is this activity important for HIF-1 induction (as has been suggested in other cell types) or are these independent pathways?*

As recommended by the editors/reviewers, we provide new data demonstrating that NFkB is not requited for the HIF-1α induction in EC under DF but NFkB plays a role in HIF1α-induced EC inflammation. We verified in our HAECs that DF increases phospho-p65 relative to UF (Figure 6). We also showed in our porcine samples that phospho-p65 is increased in the aortic arch compared with the descending thoracic aorta (Figure 7), consistent with a previous study (4). To assess whether NFkB is capable of inducing HIF-1α under DF, we subjected HAECs to NEMO binding domain peptide (NBD), a well-established NFkB inhibitor (5), and blotted for HIF-1α. As shown in Figure 6—figure supplement 4, there is no reduction in the stabilization of HIF-1α under disturbed flow. In order to further explore if HIF-1α is upstream of NFkB activation, we used either control siRNA to siRNA targeted to HIF-1α and exposed these cells to disturbed flow for 48 hours. We found that by knocking down HIF-1α, we could reduce the amount of phospho-p65 detected by Western Blotting (Figure 6). Using our mutated HIF-1α overexpression system, we pre-incubated HAECs with NEMO binding domain peptide (NBD) before treatment of HAECs with mRNA of HIF-1α for 6 hours. Treatment with NBD prevented the increase of inflammatory gene expression as the result of HIF-1α overexpression, further suggesting the HIF-1α is upstream of NFkB activation (Figure 6). Together, these data indicate that NFkB, in HAECs, is downstream of HIF-1α and disturbed flow, and that NFkB activation is not required for the HIF-1α induction in ECs under disturbed flow.

Changes were made in the section “Disturbed flow-induced metabolic reprogramming is required for endothelial activation”.

*The manuscript would also profit from additional evidence indicating that the NOX4-ROS axis leads to significant HIF1 stabilization and activation of the HIF-1 pathway in vivo. The authors could consider examining Hif-1 staining, Hif-1 reporter mice, which are available (http://www.pnas.org/content/103/1/105.full), or mice with endothelial deletion of Hif-1 (Covarrubias, Aksoylar and Horng, 2015).*

As suggested by the editors/reviewers, we have examined HIF-1α staining in mouse endothelial cells exposed to in vivo DF (inner curvatures of aortic arch) and UF (descending thoracic aorta). We performed in vivofixation of mouse (wild-type B6) aorta and stained for HIF-1α and found that endothelial HIF1α is increased in the inner curvature of aortic arch when compared to those in nearby descending thoracic artery within the same mouse (Figure 7—figure supplement 1). In addition, NOX4 is also increased in mouse EC exposed to in vivo DF (Figure 7—figure supplement 1). These results from small animal model are consistent with the measurements in large animals showing HIF1α and NOX4 are increased in EC subjected to DF in vivo (the porcine data in Figure 7). In the revised manuscript, we also point to recent evidence from Akhtar et al. (Covarrubias, Aksoylar and Horng, 2015 mentioned by the reviewers) showing partial ligation-induced disturbed flow increases EC expression of HIF1α in *apoe^-/-^* mice (6).

References

1) Chappell DC, Varner SE, Nerem RM, Medford RM, and Alexander RW. Oscillatory shear stress stimulates adhesion molecule expression in cultured human endothelium. Circ Res. 1998;82(5):532-9.

2) Morigi M, Zoja C, Figliuzzi M, Foppolo M, Micheletti G, Bontempelli M, Saronni M, Remuzzi G, and Remuzzi A. Fluid shear stress modulates surface expression of adhesion molecules by endothelial cells. Blood. 1995;85(7):1696-703.

3) Ibsen KH. The Crabtree effect: a review. Cancer Res. 1961;21(829-41.

4) Fang Y, Shi C, Manduchi E, Civelek M, and Davies PF. MicroRNA-10a regulation of proinflammatory phenotype in athero-susceptible endothelium in vivo and in vitro. Proc Natl Acad Sci U S A. 2010;107(30):13450-5.

5) May MJ, Marienfeld RB, and Ghosh S. Characterization of the Ikappa B-kinase NEMO binding domain. J Biol Chem. 2002;277(48):45992-6000.

6) Akhtar S, Hartmann P, Karshovska E, Rinderknecht FA, Subramanian P, Gremse F, Grommes J, Jacobs M, Kiessling F, Weber C, et al. Endothelial Hypoxia-Inducible Factor-1alpha Promotes Atherosclerosis and Monocyte Recruitment by Upregulating MicroRNA-19a. Hypertension. 2015;66(6):1220-6.